# Rethinking Benign Overfitting in Two-Layer Neural Networks

**Ruichen Xu** [1]    **Kexin Chen** [2]

## Abstract

Recent theoretical studies (Kou et al., 2023; Cao et al., 2022) revealed a sharp phase transition from benign to harmful overfitting when the noise-to-feature ratio exceeds a threshold—a situation common in long-tailed data distributions where atypical data is prevalent. However, such harmful overfitting rarely happens in overparameterized neural networks. Further experimental results suggested that memorization is necessary for achieving near-optimal generalization error in long-tailed data distributions (Feldman & Zhang, 2020). We argue that this discrepancy between theoretical predictions and empirical observations arises because previous feature-noise data models overlook the heterogeneous nature of noise across different data classes. In this paper, we refine the feature-noise data model by incorporating class-dependent heterogeneous noise and re-examine the overfitting phenomenon in neural networks. Through a comprehensive analysis of the training dynamics, we establish test loss bounds for the refined model. Our findings reveal that neural networks can leverage "data noise" to learn implicit features that improve the classification accuracy for long-tailed data. Our analysis also provides a training-free metric for evaluating data influence on test performance. Experimental validation on both synthetic and real-world datasets supports our theoretical results.

## 1. Introduction

Overfitting, also known as memorization, had long been considered detrimental to model generalization performance (Hastie et al., 2009). However, with the advent of over-parameterized neural networks, models can perfectly fit the training data while still exhibiting improved generalization as model complexity increases. When and why this *benign overfitting* phenomenon happens garnered significant interest within the learning theory community. Recent works, e.g., Frei et al. (2022); Cao et al. (2022); Kou et al. (2023), showed a sharp phase transition between benign and harmful overfitting in two-layer neural networks with a feature-noise data model (Allen-Zhu & Li, 2020), which assumes data is composed of a feature vector as its mean and a random Gaussian vector as its data-specific noise. Specifically, when the magnitude of the noise exceeds a threshold, neural networks memorize the data noise, leading to harmful overfitting. Nevertheless, such harmful overfitting is rarely observed in modern over-parameterized neural networks.

Empirical evidence (Feldman & Zhang, 2020; Hartley & Tsaftaris, 2022; Wang et al., 2024; Garg & Roy, 2023) indicates that memorization can, in fact, enhance generalization, especially in long-tailed data distributions characterized by substantial data-specific noise. These findings contradict current theories of the phase transition to harmful overfitting. Consequently, the following problem remains open:

*How can we theoretically explain benign overfitting in overparameterized neural networks?*

Inspired by the heterogeneous intra-class distributions in real-world datasets (as shown in Figure 1), we refine the feature-noise data model by incorporating class-dependent noise and re-examine the benign overfitting phenomenon in two-layer ReLU convolutional neural networks (CNNs). In this paper, we make the following contributions:

- We establish an enhanced feature-noise model by considering heterogeneous noise across classes. Our results with this model theoretically explain how memorization of long-tailed data boosts model performance, which cannot be predicted by existing theoretical frameworks for neural networks.

- We derive general theoretical phase transition results between benign and harmful overfitting by analyzing the test error bounds of the refined model. Our results demonstrate that although the trained neural networks

[1]Department of Information Engineering, The Chinese University of Hong Kong, Hong Kong, China [2]School of Science and Engineering, The Chinese University of Hong Kong, Shenzhen, China. Correspondence to: Ruichen Xu <rcxu642@gmail.com>.

*Proceedings of the 42ⁿᵈ International Conference on Machine Learning*, Vancouver, Canada. PMLR 267, 2025. Copyright 2025 by the author(s).

can classify data through explicit features, they can additionally utilize implicit features learned through memorization of class-dependent noise to classify long-tailed data. The findings are well-supported by real-world datasets. Our results also show a probably counterintuitive result that a class's classification accuracy for long-tailed data may decrease with the dataset sizes of other classes, giving an explanation to the observations in Sagawa et al. (2020).

- We derive new proof techniques to tackle the randomness involved in feature learning. Unlike explicit feature learning, which exhibits stable activation states and magnitudes, data noise has random activation states and magnitudes. Specifically, to tackle the random activation states, we explore the singular value distributions of neural networks to characterize the variability. Moreover, we demonstrate that the output strength of neurons randomly activated by data noise is influenced by intra-class covariance matrices.

- Our analysis provides a simple training-free metric for evaluating data memorization, unlike previous metrics that rely on training or storing multiple models. The data with high scores on our metric correspond to visually atypical samples, which are memorized to benefit model generalization, aligning with Feldman & Zhang (2020); Garg & Roy (2023).

## 1.1. Related Work

We review the topics of empirical observations and theoretical studies of memorization.

**Empirical observations of memorization.** A lot of recent empirical studies showed that memorization inevitably happens in modern over-parameterized neural networks. For instance, Feldman & Zhang (2020); Garg & Roy (2023) studied the memorization of training examples and found that neural networks tend to memorize visually atypical examples, i.e., those are rich in data-specific information. These observations motivate us to study the impact of data-specific information on benign overfitting. Our results provide a theoretical justification for these empirical observations in neural networks.

**Theoretical analyses for memorization.** A body of work theoretically examined memorization within classical machine learning frameworks, demonstrating its significance for achieving near-optimal generalization performance across various contexts, including high-dimensional linear regression (Cheng et al., 2022), prediction models (Brown et al., 2021), and generalization on discrete distributions and mixture models (Feldman, 2020). This line of

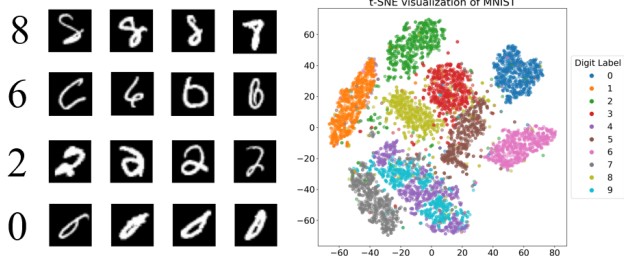

*Figure 1.* **Left:** Examples of MNIST long-tailed atypical data, which tend to be memorized by neural networks (Feldman & Zhang, 2020). **Right:** t-SNE visualization of MNIST. The heterogeneous shapes imply real-world datasets, such as MNIST, practically show heterogeneously correlated structures, providing experimental evidence for our theory.

research failed to explain learning dynamics and generalization performance in non-convex and non-smooth neural networks.

Another line of work theoretically studied memorization in neural networks by analyzing the feature learning process during training, providing analytical frameworks that extend beyond the neural tangent kernel (NTK) regime (Jacot et al., 2018; Allen-Zhu et al., 2019; Du et al., 2018). For example, Cao et al. (2022) and Kou et al. (2023) explored benign overfitting in two-layer neural networks using the feature-noise data model with homogeneous data noise distributions and showed a sharp phase transition between benign and harmful overfitting. However, they all assumed that the data-specific noise is homogeneous, leading to the conclusion that it is harmful to memorize data-specific noise. They thus fail to explain the empirical observations with long-tailed data.

## 1.2. Notation

We use lowercase letters, lowercase boldface letters, and uppercase boldface letters to denote scalars, vectors, and matrices, respectively. We use $[m]$ to denote the set $\{1, \cdots, m\}$. Given two sequences $\{x_n\}$ and $\{y_n\}$, we denote $x_n = \mathcal{O}(y_n)$ if $|x_n| \leq C_1|y_n|$ for some positive constant $C_1$ and $x_n = \Omega(y_n)$ if $|x_n| \geq C_2|y_n|$ for some positive constant $C_2$. We use $x_n = \Theta(y_n)$ if $x_n = \mathcal{O}(y_n)$ and $x_n = \Omega(y_n)$ both hold. We use $\tilde{\mathcal{O}}(\cdot), \tilde{\Theta}(\cdot)$, and $\tilde{\Omega}(\cdot)$ to hide the logarithmic factors in these notations. Given a matrix $\mathbf{A}$, we use $\|\mathbf{A}\|_F$ to denote its Frobenius norm, $\|\mathbf{A}\|_{\text{op}}$ to denote its operator norm, $\text{Tr}(\mathbf{A})$ to denote its trace, $\lambda_i(\mathbf{A})$ to denote its $i^{th}$ largest singular value, $\lambda_{\max}^+(\mathbf{A})$ and $\lambda_{\min}^+(\mathbf{A})$ to denote its maximum and minimum absolute values of non-zero singular values, and $\text{rank}(\mathbf{A})$ to denote its rank. We use the notation $(\mathbf{x}, y) \sim \mathcal{D}$ to denote that the data sample $(\mathbf{x}, y)$ is generated from a distribution $\mathcal{D}$.

## 2. Problem Setup

In this section, we introduce the problem setup, including the data model, CNNs, and the training algorithm.

Given the evidence that real-world datasets have heterogeneous class-dependent structures (as shown in Figure 1), we consider a data distribution as follows.

**Data distribution.** We define a data distribution $\mathcal{D}$ that each sample $(\mathbf{x}, y)$ is generated as

1. Sample the label $y$ following a distribution $\mathcal{K}$, whose support is $[K]$ $(K = \Theta(1))$.

2. The data $\mathbf{x} = (\mathbf{x}^{(1)}, \mathbf{x}^{(2)})$, where $\mathbf{x}^{(1)}, \mathbf{x}^{(2)} \in \mathbb{R}^d$ contains either feature or class-dependent noise patch:

- Feature patch: One patch is randomly selected as the feature patch, containing a feature vector $\mathbf{u}_y \in \mathbb{R}^d$.

- Noise patch: The remaining patch $\boldsymbol{\xi}$ is generated as $\mathbf{A}_y \boldsymbol{\zeta}$, where each coordinate of $\boldsymbol{\zeta}$ is i.i.d. drawn from $\mathcal{D}_\zeta$, a symmetric $\sigma_p = \Theta(1)$ sub-Gaussian distribution[1] with variance 1, and $\mathbf{A}_y \in \mathbb{R}^{d \times d}$ satisfies $\mathbf{u}_k^\top \mathbf{A}_y = 0$, for any $k \in [K]$. [2]

Without loss of generality, we assume that the feature vectors $\mathbf{u}_1, \cdots, \mathbf{u}_K$ are orthogonal and their norms are bounded, i.e., $\langle \mathbf{u}_i, \mathbf{u}_j \rangle = 0, \|\mathbf{u}_i\|_2 = \Theta(1)$ for all $i, j \in [K]$ and $i \neq j$.

*Remark* 2.1. The sub-Gaussian distribution of noise patches is general. In practice, pixel magnitudes in the classification tasks are bounded and thus are sub-Gaussian.

**Learner model.** We consider a two-layer CNN with ReLU activation as the learner model. Given an input $\mathbf{x} = (\mathbf{x}^{(1)}, \mathbf{x}^{(2)})$, the model with weights $\mathbf{W}$ outputs a $K$-length vector $[F_1, \cdots, F_K]$ whose elements are

$$F_k(\mathbf{W}, \mathbf{x}) = \frac{1}{m} \sum_{r=1}^{m} \sum_{j=1}^{2} \sigma\left(\left\langle \mathbf{w}_{k,r}, \mathbf{x}^{(j)} \right\rangle\right), \quad (1)$$

where $\sigma(z) = \max\{0, z\}$ denotes the ReLU activation function, and $\mathbf{w}_{k,r}$ denotes the weight vector for the $r^{th}$ neuron (totally $m$ neurons) associated with $F_k(\mathbf{W}, \mathbf{x})$.

**Training objective.** Given a training dataset with $n$ samples $\mathcal{S} = \{(\mathbf{x}_i, y_i)\}_{i=1}^{n}$ drawn from the distribution $\mathcal{D}$, we

[1]Here, we simply refer an $\alpha$ sub-Gaussian variable as a sub-Gaussian variable with variance proxy $\alpha$.

[2]This condition ensures that the noise patch is orthogonal to the feature patch. Our data distribution includes the widely adopted feature-noise data distribution with homogeneous data noise (Cao et al., 2022; Zou et al., 2023; Jelassi & Li, 2022) as a special case. By setting $K = 2$ and $\mathbf{A}_k \mathbf{A}_k^\top = \mathbf{I} - \sum_{k=1}^{K} \mathbf{u}_k \mathbf{u}_k^\top / \|\mathbf{u}_k\|_2^2$ for all $k, y \in [K]$, our data distribution becomes the same as theirs.

train the neural network by minimizing the empirical risk with the cross-entropy loss:

$$\mathcal{L}_\mathcal{S}(\mathbf{W}) = \frac{1}{n} \sum_{i=1}^{n} \mathcal{L}(\mathbf{W}, \mathbf{x}_i, y_i), \quad (2)$$

where $\mathcal{L}(\mathbf{W}, \mathbf{x}, y) = -\log(\text{logit}_y(\mathbf{W}, \mathbf{x}))$, and $\text{logit}(\cdot)$ represents the output probability of the neural network:

$$\text{logit}_y(\mathbf{W}, \mathbf{x}) = \frac{\exp(F_y(\mathbf{W}, \mathbf{x}))}{\sum_{k=1}^{K} \exp(F_k(\mathbf{W}, \mathbf{x}))}. \quad (3)$$

**Initialization.** The initial weights of the neural network's parameters are generated i.i.d. from a Gaussian distribution, i.e., $\mathbf{w}_{j,r}^{(0)} \sim \mathcal{N}(\mathbf{0}, \sigma_0^2 \mathbf{I})$, for all $j \in [K], r \in [m]$.

**Training algorithm.** We train the neural network by gradient descent (GD) with a learning rate $\eta$, i.e.,

$$\mathbf{W}^{(t+1)} = \mathbf{W}^{(t)} - \frac{\eta}{n} \sum_{(\mathbf{x},y) \in \mathcal{S}} \nabla \mathcal{L}(\mathbf{W}^{(t)}, \mathbf{x}, y). \quad (4)$$

## 3. Main Results

In this section, we present our main theoretical results. We start by introducing some conditions for our theory.

**Condition 3.1.** Suppose there exists a sufficiently large constant $C$. For certain probability parameter $\delta \in (0, 1)$, the following conditions hold:

(a) To ensure that the neurons can learn the data patterns[3], for any $i, j \in [K]$, the noise patch distributions satisfy:

$$\begin{cases} \text{Tr}(\mathbf{A}_i^\top \mathbf{A}_i) \geq Cn \max\left\{\left\|\mathbf{A}_i^\top \mathbf{A}_j\right\|_F \log(n^2/\delta), \right. \\ \left. n^{1/2} \max_{i,j \in [K]} \{\left\|\mathbf{A}_i^\top \mathbf{A}_j\right\|_F^{1/2}\} \log^{1/2}(n^2/\delta)/|\mathcal{S}_i|\right\}, \\ \left\|\mathbf{A}_i^\top \mathbf{A}_j\right\|_F / \left\|\mathbf{A}_i^\top \mathbf{A}_j\right\|_{\text{op}} \geq C\sqrt{\log(K/\delta)}, \\ \left\|\mathbf{A}_i^\top \mathbf{A}_j\right\|_F \geq C^{-1} \max_{k \neq j}\{\left\|\mathbf{A}_i^\top \mathbf{A}_k\right\|_F\}. \end{cases}$$

Moreover, there exists a threshold $c' > 0$ such that $\mathbb{P}[\zeta > c'] \geq 0.4$.

(b) To ensure that the learning problem is in a sufficiently over-parameterized setting, the training dataset size $n$, network width $m$, and dimension $d$ satisfy:

$$\begin{cases} m \geq C \log(n/\delta) \max_i\{(\lambda_{\max}^+(\mathbf{A}_i))^2/(\lambda_{\min}^+(\mathbf{A}_i))^2\}, \\ n \geq C \log(m/\delta), m \geq \Omega\left(\log(n/\delta) \log(T)^2/n\sigma_0^2\right), \\ \min\{m, d, \text{rank}(\mathbf{A}_j)\} - 0.9m \geq Cn, \\ d \geq C \log(mn/\delta). \end{cases}$$

[3]Conditions (a) on the noise patches generalize the conditions on dimension $d$ in Kou et al. (2023). Setting $\mathbf{A}_k \mathbf{A}_k^\top = \mathbf{I} - \sum_{k=1}^{K} \mathbf{u}_k \mathbf{u}_k^\top / \|\mathbf{u}_k\|_2^2$ for all $k \in [K]$, Condition (a) is similar to the dimension condition in Kou et al. (2023).

(c) To ensure that gradient descent can minimize the training loss, the learning rate $\eta$ and initialization $\sigma_0$ satisfy:

$$
\begin{cases}
\eta \leq \left( C \max_{k \in [K]} \left( \|\mathbf{u}_k\|_2 + \sqrt{1.5 \mathrm{Tr}\left(\mathbf{A}_k^\top \mathbf{A}_k\right)} \right)^2 \right)^{-1}, \\
\eta \leq mn \log(T) / \max_{j \in [K]} \{ \mathrm{Tr}(\mathbf{A}_j^\top \mathbf{A}_j) \}, \\
\sigma_0 \leq C^{-1} n^{-1} \phi \cdot \\
\left( \max_{k \in [K]} \{ \sqrt{\log(Km/\delta)} \|\mathbf{u}_k\|_2, \log(Km/\delta) \|\mathbf{A}_k\|_F \} \right)^{-1},
\end{cases}
$$

where $\phi := \min_{k_1, k_2 \in [K]} \{ \|\mathbf{A}_{k_1}^\top \mathbf{A}_{k_2}\|_F, \|\mathbf{u}_{k_1}\|_2^2 \}$.

Based on Condition 3.1, we study the model convergence and generalization performance by bounding the training loss and the zero-one test loss (accuracy) of the trained model $\mathbf{W}^{(T)}$ on distribution $\mathcal{D}_k$ whose probability density function is $\mathbb{P}_{\mathcal{D}_k}[(\mathbf{x}, y)] = \mathbb{P}_{\mathcal{D}}[(\mathbf{x}, y)|y = k]$, i.e., for all $k \in [K]$,

$$
\mathcal{L}_{\mathcal{D}_k}^{0-1}(\mathbf{W}^{(T)})
$$
$$
= \mathbb{P}_{(\mathbf{x}, y) \sim \mathcal{D}_k} \left[ F_y(\mathbf{W}^{(T)}, \mathbf{x}) \neq \max_{j \in [K]} \{ F_j(\mathbf{W}^{(T)}, \mathbf{x}) \} \right].
$$

Before presenting the main theorem, we define the set of long-tailed data with respect to the trained model $\mathbf{W}^{(T)}$.

**Definition 3.2** ($L$-Long-tailed data set). The $L$-long-tailed data distribution $\mathcal{T}_j$ for each $j \in [K]$ with model $\mathbf{W}^{(T)}$ is defined as

$$
\mathbb{P}_{\mathcal{T}_j}[(\mathbf{x}, y)] =
$$
$$
\mathbb{P}_{\mathcal{D}_j} \left[ (\mathbf{x}, y) | \langle \sum_{r \in \mathcal{R}(\boldsymbol{\xi})} \mathbf{w}_{y,r}^{(T)}, \boldsymbol{\xi} \rangle \geq L \| \mathbf{A}_y^\top \sum_{r \in \mathcal{R}(\boldsymbol{\xi})} \mathbf{w}_{y,r}^{(T)} \|_2 \right],
$$

where $\mathcal{R}(\boldsymbol{\xi}) = \{ r \in [m] : \langle \mathbf{w}_{y,r}^{(T)}, \boldsymbol{\xi} \rangle > 0 \}$.

Definition 3.2 identifies data whose equivalent noise $\zeta'$ exceeds a threshold. Specifically, for data $(\mathbf{x}, y) \sim \mathcal{D}$, the inner product term $\langle \sum_{r \in \mathcal{R}(\boldsymbol{\xi})} \mathbf{w}_{y,r}^{(T)}, \boldsymbol{\xi} \rangle$ satisfies $\langle \sum_{r \in \mathcal{R}(\boldsymbol{\xi})} \mathbf{w}_{y,r}^{(T)}, \boldsymbol{\xi} \rangle = \Theta \left( \| \mathbf{A}_y^\top \sum_{r \in \mathcal{R}(\boldsymbol{\xi})} \mathbf{w}_{y,r}^{(T)} \|_2, \zeta' \right)$, where $\zeta'$ is an equivalent random sub-Gaussian variable with variance 1.

*Remark* 3.3. Classical definitions of long-tailedness often hinge on simple one-dimensional statistics (such as Zipf distribution). However, such metrics may fail to capture the nuances of high-dimensional data. For instance, for sub-Gaussian data, norms alone cannot capture which samples are "rare" or "hard" to learn due to concentration effects. To address this, we draw inspiration from prior work that uses trained models to identify "long-tailed" samples. Specifically, Feldman & Zhang (2020) defined the long-tailed data using an influence score that quantifies the train-ing loss difference resulting from the removal of a specific sample from the dataset. Garg & Roy (2023) defined the long-tailedness of data using a curvature metric that approximates a loss Hessian matrix-related quantity of trained models averaged over all training epochs. In a similar manner, our definition leverages the trained model $\mathbf{W}^{(T)}$ to map high-dimensional samples to a one-dimensional metric, allowing us to identify long-tailed data points from the perspective of the network's learned representations.

We denote $\mathcal{S}_j$ as the set containing training data with label $j$ in the training dataset $\mathcal{S}$. We present our main result in the following theorem.

**Theorem 3.4.** *For any $\epsilon > 0$ and $k \in [K]$, under Condition 3.1, there exists $T = \tilde{\mathcal{O}}(\eta^{-1} \epsilon^{-1} \max\{ n \max_{j \in [K]} \{ \mathrm{Tr}(\mathbf{A}_j^\top \mathbf{A}_j) \}, \sqrt{md}\sigma_0, \sqrt{mK} \})$, with probability at least $1 - \delta$, the following holds:*

1. *The training loss satisfies: $\mathcal{L}_{\mathcal{S}}(\mathbf{W}^{(T)}) \leq \epsilon$.*

2. *Benign overfitting:*

   (a) *(For all data) When the signal-to-noise ratio is large, i.e., $|\mathcal{S}_k|^2 \|\mathbf{u}_k\|_2^4 \geq C_1 \cdot \max_{j \neq k} \{ |\mathcal{S}_j| \|\mathbf{A}_k^\top \mathbf{A}_j\|_F^2 \}$, the zero-one test loss satisfies:*

$$
\mathcal{L}_{\mathcal{D}_k}^{0-1}(\mathbf{W}^{(T)}) \leq \sum_{j \neq k} \exp \left( -c_1 \cdot \underbrace{\frac{|\mathcal{S}_k|^2 \|\mathbf{u}_k\|_2^4}{|\mathcal{S}_j| \|\mathbf{A}_k^\top \mathbf{A}_j\|_F^2}}_{\text{signal-to-noise ratio}} \right).
$$

   (b) *(Only for long-tailed data) When the noise correlation ratio is large, i.e., $|\mathcal{S}_k| \|\mathbf{A}_k^\top \mathbf{A}_k\|_F^2 \geq C_2 \cdot \max_{j \neq k} \{ |\mathcal{S}_j| \|\mathbf{A}_k^\top \mathbf{A}_j\|_F^2 \}$, the zero-one test loss satisfies:*

$$
\mathcal{L}_{\mathcal{T}_k}^{0-1}(\mathbf{W}^{(T)})
$$
$$
\leq \sum_{j \neq k} \exp \left( -c_2 L^2 \cdot \underbrace{\frac{|\mathcal{S}_k| \|\mathbf{A}_k^\top \mathbf{A}_k\|_F^2}{|\mathcal{S}_j| \|\mathbf{A}_k^\top \mathbf{A}_j\|_F^2}}_{\text{noise correlation ratio } \Gamma_{k,j}} \right).
$$

3. *Harmful overfitting: When the signal-to-noise ratio and noise correlation ratio are small, i.e., $|\mathcal{S}_k|^2 \|\mathbf{u}_k\|_2^4 \leq C_3 \cdot \max_{j \neq k} \{ |\mathcal{S}_j| \|\mathbf{A}_k^\top \mathbf{A}_j\|_F^2 \}$ and $|\mathcal{S}_k| \|\mathbf{A}_k^\top \mathbf{A}_k\|_F^2 \leq C_3 \cdot \max_{j \neq k} \{ |\mathcal{S}_j| \|\mathbf{A}_k^\top \mathbf{A}_j\|_F^2 \}$, the zero-one test loss satisfies $\mathcal{L}_{\mathcal{D}_k}^{0-1}(\mathbf{W}^{(T)}) \geq c_3$.*

*Here, $C_1, C_2, C_3, c_1, c_2, c_3$ are some absolute constants.*

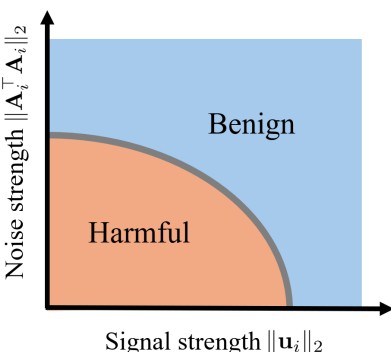

*Figure 2.* Illustration of the generalized phased transition between benign and harmful overfitting (memorization). The blue region represents a benign overfitting regime where the test loss is small. The orange region represents a harmful overfitting regime where the test loss is at a constant order. The gray band region is the setting where the test loss is not well characterized.

Theorem 3.4 characterizes a sharp phase transition between benign and harmful overfitting (memorization), visualized in Figure 2. As per Theorem 3.4, after $T$ iterations, CNNs converge to nearly optimal training loss (Statement 1). When the signal-to-noise ratio is high, the trained CNNs can achieve optimal test loss by effectively detecting explicit features $\mathbf{u}_k$ (Statement 2(a)). This result extends the previous benign overfitting results, which characterize the horizontal dimension in Figure 2. Furthermore, CNNs also classify long-tailed data by seeking to detect features learned from class-dependent noise so that the performance benefits from high noise correlation ratios (Statement 2(b)). This result at the first time characterizes the vertical dimension in Figure 2, showing that the model can also leverage the noise patch to achieve benign overfitting. Conversely, when both noise correlation ratios and signal-to-noise ratios are small, the trained CNNs incur test loss at least a constant (Statement 3). Notably, longer-tailed data (with a larger $L$) benefits more from the class-dependent noise patch.

*Remark* 3.5. By choosing $\mathbf{A}_j \mathbf{A}_j^\top = \mathbf{I} - \sum_{k=1}^{K} \mathbf{u}_k \mathbf{u}_k^\top / \|\mathbf{u}_k\|_2^2$ for all $j \in [K]$, our Statement 2(a) recovers the same convergence orders in the standard benign overfitting results (Kou et al., 2023). However, our results are more general as our results cover the whole class of sub-Gaussian data noise, $K > 2$ number of classes, and data imbalance, which is common in modern image classification tasks.

The results of Theorem 3.4 theoretically explain the following two empirical observations in neural networks for the first time.

**Long-tailed (atypical) data is important for generalization.** In Statement 2(b) of Theorem 3.4, we show that the

classification accuracy of long-tailed data increases with the noise correlation ratio. This result implies that incorporating more long-tailed data into the training dataset enhances test accuracy, providing a theoretical explanation for the empirical observation in Feldman & Zhang (2020) that including long-tailed data in the training dataset is necessary for neural networks to achieve near-optimal generalization performance.

**Increasing majority hurts minority.** Statement 2(b) of Theorem 3.4 implies that as the dataset size of class $j$, $|\mathcal{S}_j|$ increases, the upper bound on test loss of other classes $k \neq j$ increases. This leads to a possibly surprising result: The classification accuracy for long-tailed data of a specific class may decline when the sizes of other classes increase. The reason is that during training, the memorization of majority class-dependent noise (classes with more data) dominates the memorization of minority class-dependent noise. Our result theoretically explains a counter-intuitive observation in neural networks that subsampling the majority group empirically achieves low minority error (Sagawa et al., 2020).

## 4. Proof Overview

In this section, we present a proof sketch of Statement 1 and Statement 2(b) in Theorem 3.4 (Statement 2(a) uses a similar but simpler proof). Due to the space limit, we defer the complete proofs to Appendices A, B, and C.

### 4.1. Proof Sketch of Statement 1 in Theorem 3.4

We analyze the training loss in two stages. In training Stage 1, the training loss of each example decreases exponentially and stays at a constant order $\Theta(1)$. In Stage 2, the model converges to an arbitrarily small constant.

**Stage 1.** By the nature of cross-entropy, the training loss decreases exponentially, characterized by the following lemma.

**Lemma 4.1.** *Under Condition 3.1, there exists an iteration number $T_0 = \tilde{\mathcal{O}}(mn/\eta)$, so that for any $(\mathbf{x}, y) \in \mathcal{S}$ and $t \leq T_0$, the following holds:*

$$
\begin{aligned}
&\mathcal{L}(\mathbf{W}^{(t+1)}, \mathbf{x}, y) \\
&\leq \left(1 - \Theta\left(\frac{\eta}{mn}|\mathcal{S}_y|\|\mathbf{u}_y\|_2^2\right)\right)\mathcal{L}(\mathbf{W}^{(t)}, \mathbf{x}, y).
\end{aligned}
\tag{5}
$$

**Stage 2.** We show that the training loss converges to an arbitrarily small constant $\epsilon$ at a rate of $\mathcal{O}(1/T)$.

**Lemma 4.2.** *Let $\mathbf{W}^*$ be the network parameters with each neuron $\mathbf{w}_{j,r}^* = 2.5 \log\left(4(K-1)/\epsilon\right)\mathbf{u}_j/\|\mathbf{u}_j\|_2^2$. Under Condition 3.1, for any $t \in [T_0, T]$ and $(\mathbf{x}, y) \in \mathcal{S}$, the*

*following result holds:*

$$\frac{1}{t - T_0 + 1} \sum_{s=T_0}^{t} \mathcal{L}_S(\mathbf{W}^{(t)}) \leq \frac{\left\| \mathbf{W}^{(T_0)} - \mathbf{W}^* \right\|_F^2}{\eta(t - T_0 + 1)} + \frac{\epsilon}{2}.$$

After $T = \tilde{\mathcal{O}}(\eta^{-1}\epsilon^{-1}\max\{n\max_{j \in [K]}\{\mathrm{Tr}(\mathbf{A}_j^\top \mathbf{A}_j)\},$ $\sqrt{md}\sigma_0, \sqrt{mK}\})$ iterations, the training loss converges to $\epsilon$. Our analysis extends that of Kou et al. (2023)[4] by providing the convergence rate for Stage 1 and a comprehensive convergence analysis for $K$-class classification problems using cross-entropy loss.

### 4.2. Proof Sketch of Statement 2(b) in Theorem 3.4

By the definition of zero-one test loss, for any long-tailed data $(\mathbf{x}, y) \in \mathcal{T}_i, i \in [K]$, its test loss satisfies:

$$\mathcal{L}_{\mathcal{T}_i}^{0-1}(\mathbf{W}^{(T)}) \leq \sum_{j \neq i}$$

$$\mathbb{P}_{(\mathbf{x},y) \sim \mathcal{T}_i}\left[ \sum_{r=1}^{m} \sigma(\langle \mathbf{w}_{y,r}^{(T)}, \boldsymbol{\xi} \rangle) \leq \sum_{r=1}^{m} \sigma(\langle \mathbf{w}_{j,r}^{(T)}, \mathbf{u}_y \rangle) + \sigma(\langle \mathbf{w}_{j,r}^{(T)}, \boldsymbol{\xi} \rangle) \right].$$

Roughly speaking, since the neurons for a class do not learn explicit features of other classes, i.e., $\sum_{r=1}^{m} \sigma(\langle \mathbf{w}_{j,r}^{(T)}, \mathbf{u}_y \rangle)$ is small for $j \neq y$, the key to quantifying the test loss is to compare the correlation coefficients $\sum_{r=1}^{m} \sigma(\langle \mathbf{w}_{y,r}^{(T)}, \boldsymbol{\xi} \rangle)$ and $\sum_{r=1}^{m} \sigma(\langle \mathbf{w}_{j,r}^{(T)}, \boldsymbol{\xi} \rangle)$.

First, we need to quantify the number of neurons that can be activated by the random noise patch $\boldsymbol{\xi}$. We leverage the properties of the rank and singular values of the neurons corresponding to each class $j \in [K]$, i.e., $\mathbf{W}_j^{(T)} = [\mathbf{w}_{j,1}^{(T)}, \cdots, \mathbf{w}_{j,m}^{(T)}]^\top$, in the following lemma.

**Lemma 4.3.** *Under condition 3.1, the matrix $\mathbf{W}_j^{(T)}$ for all $j \in [K]$ satisfies:*

$$\begin{cases} \mathrm{rank}(\mathbf{W}_j^{(T)}\mathbf{A}_j) \geq 0.9m, \\ \lambda_1(\mathbf{W}_j^{(T)}\mathbf{A}_j) \leq \mathcal{O}(\sqrt{m}\log(T)), \\ \lambda_{\min\{m,d,\mathrm{rank}(\mathbf{A}_j)\}-n}(\mathbf{W}_j^{(T)}\mathbf{A}_j) \geq 0.1\sqrt{m}\sigma_0. \end{cases}$$

With Lemma 4.3, we can prove that the probability that $\langle \mathbf{w}_{y,r}^{(T)}, \boldsymbol{\xi} \rangle$ remains in an orthant decreases exponentially with the number of neurons $m$. Consequently, we derive the following lemma for the number of activated neurons.

**Lemma 4.4.** *Under condition 3.1, for any $j \in [K]$ and any $(\mathbf{x}, y) \sim \mathcal{D}_j$, with probability at least $1 - \exp(-\Omega(m))$, the*

---

*trained model $\mathbf{W}^{(T)}$ satisfies:*

$$\sum_{r=1}^{m} \mathbb{I}\left(\langle \mathbf{w}_{j,r}^{(T)}, \boldsymbol{\xi} \rangle\right) \geq 0.1m. \tag{6}$$

Lemma 4.4 indicates that, with high probability, at least a non-negligible amount of neurons are able to detect the noise patch of the test data.

Next, we quantify the activation magnitude of the correlation coefficients $\sum_{r=1}^{m} \sigma(\langle \mathbf{w}_{y,r}^{(T)}, \boldsymbol{\xi} \rangle)$ and $\sum_{r=1}^{m} \sigma(\langle \mathbf{w}_{j,r}^{(T)}, \boldsymbol{\xi} \rangle)$. In the following lemma, we prove that the neurons for class $j \in [K]$ mainly learn the data noise patch from class $j$.

**Lemma 4.5.** *Under Condition 3.1, for any $j \in [K], l \in [K] \backslash \{j\}, (\mathbf{x}_q, y_q) \in \mathcal{S}_j, (\mathbf{x}_a, y_a) \in \mathcal{S}_l$, existing a sufficiently large constant $C' > 0$, we have:*

$$\sum_{r=1}^{m} \sigma(\langle \mathbf{w}_{j,r}^{(T)}, \boldsymbol{\xi}_q \rangle) \geq C' \sum_{r=1}^{m} \sigma(\langle \mathbf{w}_{j,r}^{(T)}, \boldsymbol{\xi}_a \rangle). \tag{7}$$

As a result, the order of activation magnitudes is controlled by the intra-class correlations and inter-class correlations, quantified by $\|\mathbf{A}_y^\top \mathbf{A}_y\|_F$ and $\|\mathbf{A}_y^\top \mathbf{A}_j\|_F$.

**Lemma 4.6.** *Under Condition 3.1, for any $(\mathbf{x}, y) \sim \mathcal{T}_i$ and $i \in [K], j \in [K] \backslash \{y\}$, the following holds:*

$$\sum_{r=1}^{m} \sigma\left(\langle \mathbf{w}_{y,r}^{(T)}, \boldsymbol{\xi} \rangle\right) = \Theta\left(\frac{\eta m}{n} \left\|\mathbf{A}_y^\top \mathbf{A}_y\right\|_F Lb(\mathcal{S}_y)\right),$$

$$\sum_{r=1}^{m} \sigma\left(\langle \mathbf{w}_{j,r}^{(T)}, \boldsymbol{\xi} \rangle\right) = \mathcal{O}\left(\frac{\eta m}{n} \left\|\mathbf{A}_y^\top \mathbf{A}_j\right\|_F b(\mathcal{S}_j)|\zeta_1|\right),$$

*where $b(\cdot)$ is a function defined as:*

$$b(\mathcal{A}) = \sqrt{\sum_{(\mathbf{x},y) \in \mathcal{A}} \left(\sum_{t'=0}^{T-1} (1 - \mathrm{logit}_y(\mathbf{W}^{(t')}, \mathbf{x}))\right)^2}, \tag{8}$$

*and $\zeta_1$ is a sub-Gaussian variable with variance 1.*

Finally, to compare $b(\mathcal{S}_y)$ and $b(\mathcal{S}_j)$, we prove that the loss gradient coefficients are balanced.

**Lemma 4.7.** *For all $t \in [T]$ and for any $i, j \in [n]$, there exists a positive constant $\kappa$ such that:*

$$\frac{1 - \mathrm{logit}_{y_i}(\mathbf{W}^{(t)}, \mathbf{x}_i)}{1 - \mathrm{logit}_{y_j}(\mathbf{W}^{(t)}, \mathbf{x}_j)} \leq \kappa. \tag{9}$$

Leveraging Hoeffding's inequality, Lemma 4.6, and 4.7, we derive Statement 2(b) in Theorem 3.4.

---

[4]Kou et al. (2023) provided convergence analysis for training loss in two-layer CNNs with binary classification and logistic loss.

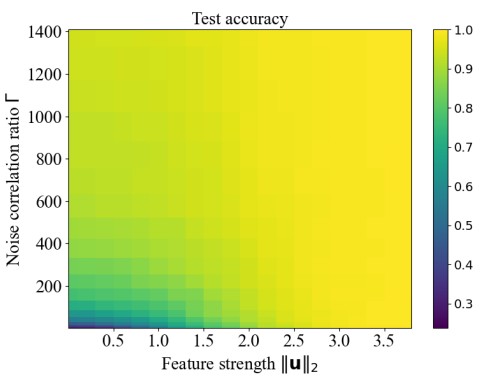
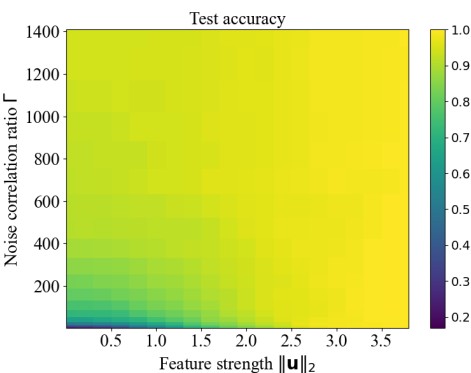

(a) Test accuracy heatmap under Gaussian data noise. Each entry of $\zeta$ follows $\mathcal{N}(0, 1)$.

(b) Test accuracy heatmap under uniform data noise. Each entry of $\zeta$ follows $\mathcal{U}(-\sqrt{3}, \sqrt{3})$.

*Figure 3.* Impacts of feature strength and noise correlation ratio on test accuracy.

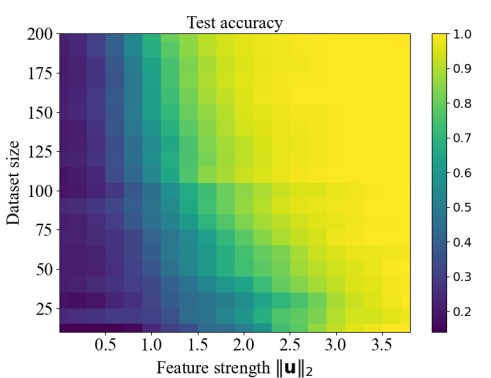
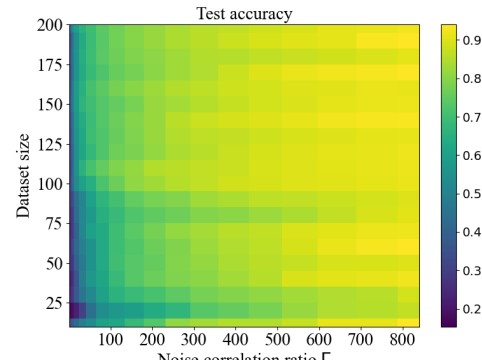

(a) Test accuracy heatmap with feature strength and dataset size. Each entry of $\zeta$ follows $\mathcal{N}(0, 1)$.

(b) Test accuracy heatmap with noise correlation ratio and dataset size. Each entry of $\zeta$ follows $\mathcal{N}(0, 1)$.

*Figure 4.* Impacts of data size on test accuracy.

## 5. Experiments

In this section, we first validate our theory in Theorem 3.4 by constructing datasets and models following our problem setup in Section 2. We further verify our conclusions with real-world datasets MNIST (LeCun et al., 1998), CIFAR-10, and CIFAR-100 (Krizhevsky et al., 2009).

### 5.1. Synthetic Datasets

**Synthetic data generation.** We generate a synthetic dataset following the data distribution outlined in Section 2. The dataset comprises a total of $K = 5$ classes with a dimensionality of $d = 1000$. We set the feature patches for each $k \in [K]$ as $\mathbf{u}_k = \mathbf{U} \cdot \|\mathbf{u}_k\|_2 \cdot \mathbf{z}_k$, where $\mathbf{z}_k$ keeps the $k^{th}$ entry to be one and other entries zeros, and $\mathbf{U}$ is a randomly generated unitary matrix. Without loss of generality, we consider $\mathbf{A}_k$ to be matrices with all but one non-zero eigenvalue fixed as 0.5. Then, we tune the non-fixed non-zero eigenvalue to change the noise correlation ratio

accordingly.

**Setup.** We train a two-layer neural network as defined in Section 2 with $m = 100$ neurons. We use the default initialization method in PyTorch to initialize the neurons' parameters. We train the neural networks using GD with a learning rate $\eta = 0.05$ over 20 epochs. To explore the effects of feature strength $\|\mathbf{u}_i\|_2$, noise correlation ratio $\Gamma_{i,j}$, and dataset size $|\mathcal{S}_i|$, we simply fix $\|\mathbf{u}_i\|_2 = \|\mathbf{u}\|_2$, $\Gamma_{i,j} = \Gamma$, and $|\mathcal{S}_i| = s$, where $\|\mathbf{u}\|_2$, $\Gamma$, and $s$ are tunable parameters for all class $i, j \in [K]$ ($j \neq i$).

**Effects of noise correlation and feature strength.** We simulate with feature strength $\|\mathbf{u}\|_2$ ranging from 0.001 to 3.8 and noise correlation ratio $\Gamma$ ranging from 1 to 1400. We fix the dataset size of each class as $|\mathcal{S}_i| = 100$, for all $i \in [K]$. The resulting heatmap of test accuracy in relation to data feature size and noise correlation is presented in Figure 3. As illustrated, the test loss decreases not only with increasing feature strength but also with higher noise

Influence
score

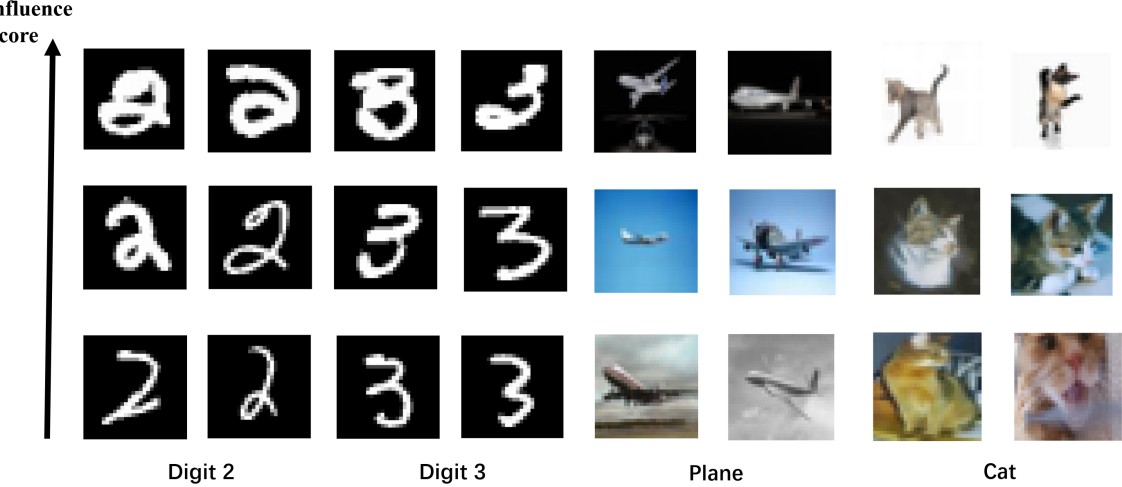

Digit 2          Digit 3          Plane          Cat

*Figure 5.* Visualization of images with different influence scores.

correlation ratios. Notably, with high noise correlation ratios, neural networks can achieve near-optimal test accuracy even when feature strength approaches nearly zero.

**Effect of dataset size.** We verify the impact of the dataset sizes in the two types of benign overfitting in Statements 2(a) and 2(b) of Theorem 3.4. We fix the noise correlation ratio as $\Gamma = 1.2$ with $\zeta$ following Gaussian distribution and vary the feature size ranging from $0.01$ to $4$ and dataset size ranging from $10$ to $200$ to explore the effect of dataset size versus feature strength on test accuracy (Figure 4a). We fix the feature strength as $\|\mathbf{u}\|_2 = 1e - 6$ and tune noise correlation ratio $\Gamma$ (with $\zeta$ following Gaussian distribution) ranging from $1$ to $800$ and dataset size ranging from $10$ to $200$ to explore the effect of dataset size versus noise correlation ratio (Figure 4b). These results verify our theory. Specifically, for classification utilizing explicit features (Figure 4a), the test accuracy increases with increasing dataset size, matching our Statement 2(a) in Theorem 3.4. In contrast, for classification utilizing implicit features (Figure 4b), the test accuracy remains in a constant order with the dataset size, matching Statement 2(b) in Theorem 3.4.

### 5.2. Real-World Datasets

We verify the effects of noise correlation on MNIST (Le-Cun et al., 1998), CIFAR-10, and CIFAR-100 (Krizhevsky et al., 2009).

**Noise correlation ratio verifications.** To verify our results in Theorem 3.4, we compute the squared Frobenius norm, i.e., $\|\mathbf{A}_i^\top \mathbf{A}_j\|_F^2$, among different classes of MNIST. We provide the computation details in Appendix E. As shown in Figure 6, real datasets such as MNIST indeed

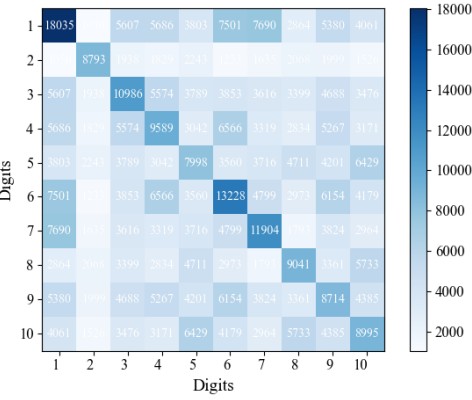

*Figure 6.* Visualization of the squared Frobenius norm among classes in MNIST.

have large noise correlation ratios, satisfying the conditions for long-tailed data benign overfitting predicted in our theory (Statement 2(b) of Theorem 3.4).

**Data influence score.** We quantify the impact of a single data sample $(\mathbf{x}, y)$ by measuring its impact on $\|\mathbf{A}_y^\top \mathbf{A}_y\|_F^2$ (A determining factor of the long-tailed data loss as in Statement 2(b) of Theorem 3.4). We observe that $\|\mathbf{A}_y^\top \mathbf{A}_y\|_F$ is, in fact, the same as the Frobenius norm of the covariance matrix of class $y$'s distribution and hence we consider an influence score of an image $(\mathbf{x}, y) \in \mathcal{S}$ as:

$$\text{infl}(\mathbf{x}) = \left\|\hat{\Sigma}(\mathcal{S}_y)\right\|_F^2 - \left\|\hat{\Sigma}(\mathcal{S}_y \backslash \{\mathbf{x}\})\right\|_F^2, \quad (10)$$

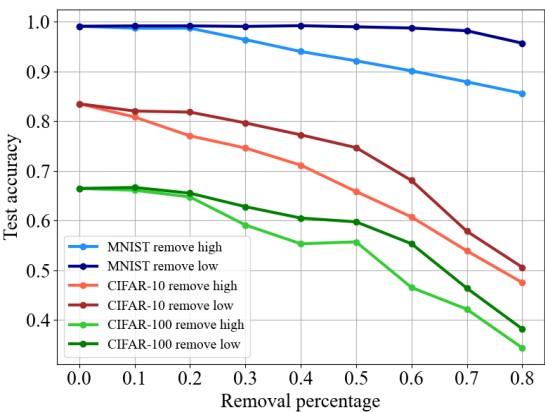

*Figure 7.* Effect on the test accuracy of removing a fraction of examples with high and low influence scores.

where $\hat{\Sigma}(\mathcal{S}_y)$ denotes the estimated covariance of the underlying data distribution within the dataset $\mathcal{S}_y$.[5]

**What influence scores indicate.** We visualize the data with high and low influence scores in Figure 5. The data with high influence scores are atypical data, which can be interpreted by scrawly written digits and hard-to-classify objects in the experimental datasets.

**Inspiration from influence scores.** We sort the data from MNIST, CIFAR-10, and CIFAR-100 based on the influence score defined in Eq. (10). In Figure 7, we remove a portion of data with high and low influence scores and train the remaining data on LeNet (LeCun et al., 1998), ResNet-18, and ResNet-101 (He et al., 2016) to assess their impact on model accuracy. We observe that the accuracy drop of removing data with high influence scores is significantly larger than that of removing data with low influence scores. This observation verifies our theory (Statement 2(b) in Theorem 3.4) that reducing the squared Frobenius norm significantly hurts the model's test performance as it reduces accuracy on long-tailed data.

## 6. Conclusions

In this paper, we re-examine the overfitting phenomenon in overparameterized neural networks. Specifically, we enhance the widely used feature-noise data model by incorporating heterogeneous noise across classes. Our findings reveal that neural networks can learn implicit features from class-dependent noise and leverage these features to enhance classification accuracy in long-tailed data distributions. These findings align with the practical observations that memorization can enhance generalization performance. Furthermore, our analysis provides a sim-

---

[5]We estimate the covariance matrix using the sample covariance matrix.

ple training-free metric for evaluating data influence on test performance. Experiments on both synthetic and real-world datasets further validate our theory.

## Impact Statement

Benign overfitting is one of the most fundamental phenomena for neural networks. Our work discovers a new mechanism of benign overfitting, providing new insights not only on theoretical studies but also on improving training pipelines, such as designing data collections to achieve desired properties. Additionally, we proposed a training-free metric to evaluate the influence of data on test performance.

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

# A. Additional Related Work

**Implicit bias in neural networks.** A line of studies found that training algorithms of neural networks prefer solutions with certain properties, such as flatness (Damian et al., 2021), sparsity (HaoChen et al., 2021), max margin (Soudry et al., 2018), and bounded $\ell_\infty$ norm (Xie & Li, 2024), leading to better generalization. In contrast, this paper focuses on studying the training dynamics, which detailedly characterizes how neural networks learn features to boost generalization.

# B. Key lemmas

In this section, we present some important lemmas that illustrate the key properties of the data and neural networks.

**Lemma B.1.** *Let $\mathbf{Z} \in \mathbb{R}^{m \times n}(m > n)$ be a matrix whose entries are independent and identically distributed Gaussian variables, i.e., $\mathbf{Z}_{i,j} \sim \mathcal{N}(0,1)$ for all $i \in [m], j \in [n]$. With probability at least $1 - \delta$, all singular values of $\mathbf{Z}$, $\lambda_i(\mathbf{Z})$, for all $i \in [n]$ satisfies*

$$\lambda_i(\mathbf{Z}) \geq \sqrt{m} - 2\sqrt{8\log(\frac{2}{\delta}) + 8\log(9)n}. \tag{11}$$

Lemma B.1 follows from the concentration inequality of Gaussian random matrices (Vershynin, 2018).

**Lemma B.2.** *For $n$ random $\sigma_p$ sub-Gaussian variable $x_1, \cdots, x_n$, with probability $1 - \delta$, we have*

$$\mathbb{P}[|x| \geq t] \leq 2\exp\left(-\frac{t^2}{2\sigma_p^2}\right). \tag{12}$$

*Proof.* Based on the definition of sub-Gaussian distribution, with probability of $1 - \delta/n$, we have

$$|x_i| \geq \sqrt{2\sigma_p^2 \log(\frac{2n}{\delta})}. \tag{13}$$

By Union bound, we finishes the proof. □

**Lemma B.3.** *Suppose two zero-mean random vectors $\mathbf{x}, \mathbf{y} \in \mathbb{R}^d$ are generated as $\mathbf{x} = \mathbf{A}\boldsymbol{\zeta}_1, \mathbf{y} = \mathbf{A}\boldsymbol{\zeta}_2$, where $\boldsymbol{\zeta}_i$'s each coordinate is independent, symmetric, and $\sigma_p$ sub-Gaussian with $\mathbb{E}[\zeta_{i,j}^2] = 1$, for any $i \in [2], j \in [d]$. Then, $\mathbf{x}$ and $\mathbf{y}$ satisfy*

$$\mathbb{P}\left[|\langle \mathbf{x}, \mathbf{y} \rangle| \geq t\right] \leq 2\exp\left(-\Omega\left(\min\left\{\frac{t^2}{\|\mathbf{A}^\top \mathbf{A}\|_F^2 \sigma_p^4}, \frac{t}{\|\mathbf{A}^\top \mathbf{A}\|_{\text{op}} \sigma_p^2}\right\}\right)\right). \tag{14}$$

*Proof.* We have

$$\langle \mathbf{x}, \mathbf{y} \rangle = \langle \mathbf{A}\boldsymbol{\zeta}_1, \mathbf{A}\boldsymbol{\zeta}_2 \rangle = \boldsymbol{\zeta}_1^\top \mathbf{A}^\top \mathbf{A}\boldsymbol{\zeta}_2 = \boldsymbol{\zeta}_1^\top \mathbf{U}^\top \Lambda \mathbf{U}\boldsymbol{\zeta}_2 = \tilde{\boldsymbol{\zeta}}_1^\top \Lambda \tilde{\boldsymbol{\zeta}}_2 = \text{Tr}\left(\Lambda \tilde{\boldsymbol{\zeta}}_2 \tilde{\boldsymbol{\zeta}}_1^\top\right). \tag{15}$$

As $\tilde{\boldsymbol{\zeta}}_1$ and $\tilde{\boldsymbol{\zeta}}_2$ are isotropic, by Bernstein's inequality, we have

$$\mathbb{P}\left[\text{Tr}\left(\Lambda \tilde{\boldsymbol{\zeta}}_2 \tilde{\boldsymbol{\zeta}}_1^\top\right) \geq t\right] \leq 2\exp\left(-\Omega\left(\min\left\{\frac{t^2}{\|\mathbf{A}^\top \mathbf{A}\|_F^2 \sigma_p^4}, \frac{t}{\|\mathbf{A}^\top \mathbf{A}\|_{\text{op}} \sigma_p^2}\right\}\right)\right). \tag{16}$$

□

**Lemma B.4.** *Suppose a zero-mean random vector $\mathbf{x} \in \mathbb{R}^d$ is generated as $\mathbf{x} = \mathbf{A}\boldsymbol{\zeta}$, where $\boldsymbol{\zeta}$'s each coordinate is independent, symmetric, and $\sigma_p$ sub-Gaussian with $\mathbb{E}\left[\zeta_j^2\right] = 1$, for any $j \in [d]$. Then, $\mathbf{x}$ satisfies*

$$\mathbb{P}\left[\|\mathbf{x}\|_2^2 - \text{Tr}\left(\mathbf{A}^\top \mathbf{A}\right) \geq t\right] \leq 2\exp\left(-\Omega\left(\min\left\{\frac{t^2}{\|\mathbf{A}^\top \mathbf{A}\|_F^2 \sigma_p^4}, \frac{t}{\|\mathbf{A}^\top \mathbf{A}\|_{\text{op}} \sigma_p^2}\right\}\right)\right). \tag{17}$$

*Proof.* We have

$$\|\mathbf{x}\|_2^2 = \langle \mathbf{A}\boldsymbol{\zeta}_1, \mathbf{A}\boldsymbol{\zeta}_1 \rangle = \boldsymbol{\zeta}_1^\top \mathbf{A}^\top \mathbf{A}\boldsymbol{\zeta}_1 = \mathrm{Tr}\left(\mathbf{A}^\top \mathbf{A}\boldsymbol{\zeta}_1\boldsymbol{\zeta}_1^\top\right). \tag{18}$$

Then, expectation of $\|\mathbf{x}\|_2$ satisfies

$$\mathbb{E}\left[\|\mathbf{x}\|_2^2\right] = \mathrm{Tr}\left(\mathbf{A}^\top \mathbf{A}\right). \tag{19}$$

By Bernstein's inequality, we have

$$\mathbb{P}\left[\|\mathbf{x}\|_2^2 - \mathrm{Tr}\left(\mathbf{A}^\top \mathbf{A}\right) \geq t\right] \leq 2\exp\left(-\Omega\left(\min\left\{\frac{t^2}{\|\mathbf{A}^\top \mathbf{A}\|_F^2 \, \sigma_p^4}, \frac{t}{\|\mathbf{A}^\top \mathbf{A}\|_{\mathrm{op}} \, \sigma_p^2}\right\}\right)\right). \tag{20}$$

$\square$

**Lemma B.5.** *Suppose two zero-mean random vectors $\mathbf{x}_1, \mathbf{x}_2 \in \mathbb{R}^d$ are generated as $\mathbf{x}_1 = \mathbf{A}\boldsymbol{\zeta}_1, \mathbf{x}_2 = \mathbf{B}\boldsymbol{\zeta}_2$, where $\boldsymbol{\zeta}_i$'s each coordinate is independent, symmetric, and $\sigma_p$ sub-Gaussian with $\mathbb{E}[\zeta_{i,j}^2] = 1$, for any $i \in [2], j \in [d]$. Then, $\mathbf{x}_1$ and $\mathbf{x}_2$ satisfy*

$$\mathbb{P}\left[|\langle \mathbf{x}_1, \mathbf{x}_2\rangle| \geq t\right] \leq 2\exp\left(-\Omega\left(\min\left\{\frac{t^2}{\|\mathbf{A}^\top \mathbf{B}\|_F^2}, \frac{t}{\|\mathbf{A}^\top \mathbf{B}\|_{\mathrm{op}}}\right\}\right)\right). \tag{21}$$

*Proof.* We have

$$\langle \mathbf{x}, \mathbf{y}\rangle = \langle \mathbf{A}\boldsymbol{\zeta}_1, \mathbf{B}\boldsymbol{\zeta}_2 \rangle = \boldsymbol{\zeta}_1^\top \mathbf{A}^\top \mathbf{B}\boldsymbol{\zeta}_2 = \mathrm{Tr}\left(\mathbf{A}^\top \mathbf{B}\boldsymbol{\zeta}_2\boldsymbol{\zeta}_1^\top\right). \tag{22}$$

Then, by Bernstein's inequality, we have

$$\begin{aligned}
\mathbb{P}\left[|\langle \mathbf{x}_1, \mathbf{x}_2\rangle| \geq t\right] &= \mathbb{P}\left[\mathrm{Tr}\left(\mathbf{A}^\top \mathbf{B}\boldsymbol{\zeta}_2\boldsymbol{\zeta}_1^\top\right) \geq t\right] \\
&\leq 2\exp\left(-\Omega\left(\min\left\{\frac{t^2}{\|\mathbf{A}^\top \mathbf{B}\|_F^2 \, \sigma_p^4}, \frac{t}{\|\mathbf{A}^\top \mathbf{B}\|_{\mathrm{op}} \, \sigma_p^2}\right\}\right)\right).
\end{aligned} \tag{23}$$

$\square$

**Lemma B.6.** *Let $x_1, \cdots, x_m$ be $m$ independent zero-mean Gaussian variables. Denote $z_i$ as indicators for signs of $x_i$, i.e., for all $i \in [m]$,*

$$z_i = \begin{cases} 1, & x_i > 0, \\ 0, & x_i \leq 0. \end{cases} \tag{24}$$

*Then, we have*

$$\Pr\left[\sum_{i=1}^m z_i \geq 0.4m\right] \geq 1 - \exp\left(-\frac{8}{25}m\right). \tag{25}$$

*Proof.* Because $z_i, i \in [m]$ are bounded in $[0, 1]$, $z_i, i \in [m]$ are sub-Gaussian variables. By Hoeffding's inequality, we have

$$\Pr\left[m\left(\frac{1}{m}\sum_{i=1}^m z_i\right) \leq m\left(\frac{1}{2} - \epsilon\right)\right] \leq \exp\left(-\frac{2m^2\epsilon^2}{m(1/16)}\right). \tag{26}$$

Let $\epsilon = 0.1$, we have

$$\Pr\left[\sum_{i=1}^m z_i \leq 0.4m\right] \leq \exp\left(-\frac{8}{25}m\right). \tag{27}$$

Therefore, we have

$$\Pr\left[\sum_{i=1}^{m} z_i \geq 0.4m\right] \geq 1 - \exp\left(-\frac{8}{25}m\right). \tag{28}$$

This completes the proof. $\square$

**Lemma B.7.** *For any constant $t \in (0, 1]$ and $x \in [-a, b], a, b > 0$, we have*

$$\log(1 + t(\exp(x) - 1)) \leq \Gamma(x)x, \tag{29}$$

*where $\Gamma(x) = \mathbb{I}(x \geq 0) + [\log(1 + t(\exp(-a) - 1))/ - a]\,\mathbb{I}(x < 0)$.*

*Proof.* First, considering $x \geq 0$, we have

$$\frac{\partial \log(1 + t(\exp(x) - 1))}{\partial t} = \frac{\exp(x) - 1}{1 + t(\exp(x) - 1)} \geq 0. \tag{30}$$

Thus, $\log(1 + t(\exp(x) - 1)) \leq x, \forall x > 0$. Second, considering $x < 0$, we have

$$\frac{\partial^2 \log(1 + t(\exp(x) - 1))}{\partial x^2} = \frac{(1 - t)t\exp(x)}{[1 + t(\exp(x) - 1)]^2} \geq 0. \tag{31}$$

So $\log(1 + t(\exp(x) - 1))$ is a convex function of $x$. We can conclude that

$$\log(1 + t(\exp(x) - 1)) \leq \frac{\log(1 + t(\exp(-a) - 1))}{-a}x, \forall x < 0. \tag{32}$$

This completes the proof. $\square$

**Lemma B.8.** *With input $\mathbf{x} \in \mathbb{R}^K$, the function $f(\mathbf{x}) = -\log(\exp(x_i)/\sum_{j \in [K]} \exp(x_j))$ with any $i \in [K]$ is convex.*

*Proof.* For any $\mathbf{x}_1, \mathbf{x}_2 \in \mathbb{R}^K$ and $\alpha \in [0, 1]$, we have

$$
\begin{aligned}
\alpha f(\mathbf{x}_1) + (1 - \alpha)f(\mathbf{x}_2) &= -\alpha \log\left(\frac{\exp(x_{1,i})}{\sum_{j \in [K]} \exp(x_{1,j})}\right) - (1 - \alpha) \log\left(\frac{\exp(x_{2,i})}{\sum_{j \in [K]} \exp(x_{2,j})}\right) \\
&= -\log\left(\left(\frac{\exp(x_{1,i})}{\sum_{j \in [K]} \exp(x_{1,j})}\right)^\alpha \left(\frac{\exp(x_{2,i})}{\sum_{j \in [K]} \exp(x_{2,j})}\right)^{1-\alpha}\right) \\
&= -\log\left(\frac{\exp(\alpha x_{1,i})}{(\sum_{j \in [K]} \exp(x_{1,j}))^\alpha} \frac{\exp((1 - \alpha)x_{2,i})}{(\sum_{j \in [K]} \exp(x_{2,j}))^{1-\alpha}}\right) \\
&= -\log\left(\frac{\exp(\alpha x_{1,i} + (1 - \alpha)x_{2,i})}{(\sum_{j \in [K]} \exp(x_{1,j}))^\alpha (\sum_{j \in [K]} \exp(x_{2,j}))^{1-\alpha}}\right) \\
&\geq -\log\left(\frac{\exp(\alpha x_{1,i} + (1 - \alpha)x_{2,i})}{\sum_{j \in [K]} \exp(\alpha x_{1,j} + (1 - \alpha)x_{2,j})}\right) \\
&= f(\alpha \mathbf{x}_1 + (1 - \alpha)\mathbf{x}_2).
\end{aligned}
\tag{33}
$$

This finishes the proof. $\square$

**Lemma B.9.** *A vector $\mathbf{z}$ uniformly sampled from $\mathbb{S}^{d-1}$ satisfies*

$$\mathbb{P}\left[\left|\frac{1}{d}\sum_{i=1}^{d} \mathbb{I}(z_i) - \frac{1}{2}\right| \geq t\right] \leq \exp\left(-2dt^2\right). \tag{34}$$

This lemma directly follows from Hoeffding's inequality.

**Lemma B.10.** *For a constant $0 < t < 1$, a $\sigma_p$ sub-Gaussian variable $x$ with variance 1 satisfies*

$$\mathbb{P}\left[|x| > t\right] \geq \Omega((1 - t^2)^2). \tag{35}$$

*Proof.* Since $x$ is a sub-Gaussian variable with variance 1, we have $\mathbb{E}[x] = 1$. Applying Paley–Zygmund inequality to $x^2$, we have

$$\mathbb{P}[x^2 \geq t] \geq (1 - t)^2 \frac{1}{\mathbb{E}[x^4]}. \tag{36}$$

As the fourth moment of sub-Gaussian variable $\mathbb{E}[x^4]$ is bounded by $\mathcal{O}(\sigma_p^4)$ (Vershynin, 2018), we have

$$\mathbb{P}[x^2 \geq t] \geq \frac{C_4}{\sigma_p^4}(1 - t)^2, \tag{37}$$

where $C_4$ is a constant. This completes the proof. $\square$

**Lemma B.11.** *Let $\mathbf{A} \in \mathbb{R}^{m \times n}$ be a matrix with rank $M$. Suppose $\delta > 0$ and $m \geq \Omega(\log(n/\delta)(\lambda_{\max}^+(\mathbf{A}))^2/((\lambda_{\min}^+(\mathbf{A}))^2))$. With probability $1 - \delta$, for any orthant $\mathcal{T} \in \mathbb{R}^m$ and vector $\mathbf{x} \in \mathbb{R}^n$ is a random sub-Gaussian vector with each coordinate follows $\mathcal{D}_\xi$, we have*

$$\mathbb{P}\left[\mathbf{A}\mathbf{x} \in \mathcal{T}\right] \leq (0.6)^M. \tag{38}$$

*Proof.* Using singular value decomposition for $\mathbf{A}$, for any orthant $\mathcal{T} \in \mathbb{R}^m$, we have

$$\mathbb{P}\left[\mathbf{A}\mathbf{x} \in \mathcal{T}\right] = \mathbb{P}\left[\mathbf{U}\boldsymbol{\Sigma}\mathbf{x} \in \mathcal{T}\right]. \tag{39}$$

Without loss of generality, we assume the orthant $\mathcal{T}$ is that with all positive entries. Then, we have

$$\mathbb{P}\left[\mathbf{A}\mathbf{x} \in \mathcal{T}\right] \leq \mathbb{P}\left[\tilde{\mathbf{U}}\tilde{\boldsymbol{\Sigma}}\mathbf{x} \in \mathcal{T}\right], \tag{40}$$

where

$$\tilde{\mathbf{U}} = \begin{bmatrix} \mathbf{1} \cdot 1/\sqrt{m} & \tilde{\mathbf{U}}_2 & \cdots & \tilde{\mathbf{U}}_m \end{bmatrix}, \tag{41}$$

and

$$\tilde{\boldsymbol{\Sigma}} = \begin{bmatrix} \lambda_{\max}^+(\mathbf{A}) & 0 & \cdots & 0 & \\ 0 & \lambda_{\min}^+(\mathbf{A}) & \cdots & 0 & \\ \vdots & \vdots & & \vdots & \cdots \\ 0 & 0 & \cdots & \lambda_{\min}^+(\mathbf{A}) & \\ & & \cdots & & 0 \end{bmatrix}, \tag{42}$$

Here, $\tilde{\boldsymbol{\Sigma}}$ is generated by replacing the non-zero singular values other than the largest one with $\lambda_{\min}^+(\mathbf{A})$. $\mathbf{1}$ is a vector with all one entries. In the following, we abbreviate $\lambda_{\max}^+(\mathbf{A})$ and $\lambda_{\min}^+(\mathbf{A})$ as $\lambda_{\max}^+$ and $\lambda_{\min}^+$ for convenience. Then, we have

$$\tilde{\mathbf{U}}\tilde{\boldsymbol{\Sigma}}\mathbf{x} = \lambda_{\max}^+ \cdot \frac{1}{\sqrt{m}} \cdot \mathbf{1} \cdot x_1 + \lambda_{\min}^+ \sum_{i=2}^{M} x_i \tilde{\mathbf{U}}_i. \tag{43}$$

Here, as each coordinate of $\mathbf{x}$ is generate from $\mathcal{D}_\xi$, by Lemma B.2, we have

$$|x_i| \leq \sqrt{2\sigma_p^2 \log(\frac{2n}{\delta})}, \tag{44}$$

with probability $1 - \delta$. Then, we have

$$
\begin{aligned}
\mathbb{P}\left[\tilde{\mathbf{U}}\tilde{\mathbf{\Sigma}}\mathbf{x} \in \mathcal{T}\right] =&\mathbb{P}\left[(\lambda_{\max}^+ - \lambda_{\min}^+) \cdot \frac{1}{\sqrt{m}} \cdot \mathbf{1} \cdot x_1 + \lambda_{\min}^+ \sum_{i=1}^M x_i \tilde{\mathbf{U}}_i \in \mathcal{T}\right] \\
\leq&\mathbb{P}\left[\lambda_{\max}^+ \cdot \frac{1}{\sqrt{m}} \cdot \mathbf{1} \cdot \sqrt{2\sigma_p^2 \log(\frac{2n}{\delta})} + \lambda_{\min}^+ \sum_{i=1}^M x_i \tilde{\mathbf{U}}_i \in \mathcal{T}\right] \\
=&\mathbb{P}\left[\mathbf{1} \cdot \frac{\sqrt{2\sigma_p^2 \log(2n/\delta)}\lambda_{\max}^+}{\sqrt{m}} + \lambda_{\min}^+ \mathbf{x} \in \mathcal{T}\right].
\end{aligned}
\tag{45}
$$

As the entries of $\mathbf{x}$ are independent, we can bound each entry independently. Then, when $m \geq \Omega(\log(n/\delta)(\lambda_{\max}^+)^2/((\lambda_{\min}^+)^2))$, we have

$$
\mathbb{P}\left[\lambda_{\min}^+ x_i + \frac{\sqrt{2\sigma_p^2 \log(2n/\delta)}\lambda_{\max}^+}{\sqrt{m}} < 0\right] \geq 0.4,
\tag{46}
$$

by the property of $\mathcal{D}_\xi$, resulting in

$$
\mathbb{P}\left[\tilde{\mathbf{U}}\tilde{\mathbf{\Sigma}}\mathbf{x} \in \mathcal{T}\right] \leq (0.6)^M.
\tag{47}
$$

This completes the proof. □

**Lemma B.12.** *Let $\mathbf{A} \in \mathbb{R}^{m \times n}$ be a matrix with rank $M$. Suppose $m \geq \Omega(\log(n/\delta)(\lambda_{\max}^+(\mathbf{A}))^2/((\lambda_{\min}^+(\mathbf{A}))^2))$. With probability $1 - \delta$, we have*

$$
\mathbb{P}\left[\sum_{i=1}^m \mathbb{I}((\mathbf{A}\mathbf{x})_i \leq 0) \geq 0.9m\right] \leq \exp(-0.07m).
\tag{48}
$$

*Proof.* By Lemma B.9, the number $N_o$ of orthants that have more than $0.9m$ negative entries satisfies

$$
N_o \leq 2^m \cdot \exp(-0.32m) = \exp((\log 2 - 0.32)m).
\tag{49}
$$

Then, by Lemma B.11, we have

$$
\begin{aligned}
\mathbb{P}\left[\sum_{i=1}^m \mathbb{I}((\mathbf{A}\mathbf{x})_i \leq 0) \geq 0.9m\right] \leq&0.6^M \cdot \exp((\log 2 - 0.32)m) \\
=&\exp(\log(0.6)M + (\log 2 - 0.32)m) \\
\leq&\exp(-0.51M + 0.38m).
\end{aligned}
\tag{50}
$$

As long as $M \geq 0.9m$, we have

$$
\mathbb{P}\left[\sum_{i=1}^m \mathbb{I}((\mathbf{A}\mathbf{x})_i \leq 0) \geq 0.9m\right] \leq \exp(-0.07m).
\tag{51}
$$

This completes the proof. □

**Lemma B.13.** *Suppose that $\delta > 0$ and $\operatorname{Tr}(\mathbf{A}_i^\top \mathbf{A}_i) = \Omega\left(\max\left\{\left(\|\mathbf{A}_i^\top \mathbf{A}_j\|_F^2 \sigma_p^4 \log(6n/\delta)\right)^{1/2}, \|\mathbf{A}_i^\top \mathbf{A}_j\|_{\text{op}} \sigma_p^2 \log(6n/\delta)\right\}\right)$. For all $i, j \in [K]$, with probability $1 - \delta$, we have*

$$
\frac{1}{2}\operatorname{Tr}(\mathbf{A}_{y_i}^\top \mathbf{A}_{y_i}) \leq \|\boldsymbol{\xi}_i\|_2^2 \leq \frac{3}{2}\operatorname{Tr}(\mathbf{A}_{y_i}^\top \mathbf{A}_{y_i}),
\tag{52}
$$

$$
|\langle \boldsymbol{\xi}_i, \boldsymbol{\xi}_j \rangle| \leq \mathcal{O}\left(\max\left\{\left(\|\mathbf{A}_{y_i}^\top \mathbf{A}_{y_j}\|_F^2 \log(\frac{6n^2}{\delta})\right)^{1/2}, \|\mathbf{A}_{y_i}^\top \mathbf{A}_{y_j}\|_{\text{op}} \log(\frac{6n^2}{\delta})\right\}\right).
\tag{53}
$$

*Proof.* By Lemma B.4, with probability at least $1 - \delta/3n$, there exists a constant $C_1 > 0$ such that

$$\|\boldsymbol{\xi}_i\|_2^2 \leq \mathrm{Tr}(\mathbf{A}_{y_i}^\top \mathbf{A}_{y_i}) + \max\left\{\left(\|\mathbf{A}_{y_i}^\top \mathbf{A}_{y_i}\|_F^2 \, \sigma_p^4 \frac{1}{C_1} \log(\frac{6n}{\delta})\right)^{1/2}, \|\mathbf{A}_{y_i}^\top \mathbf{A}_{y_i}\|_{\mathrm{op}} \sigma_p^2 \frac{1}{C_1} \log(\frac{6n}{\delta})\right\},$$

and

$$\|\boldsymbol{\xi}_{y_i}\|_2^2 \geq \mathrm{Tr}(\mathbf{A}_{y_i}^\top \mathbf{A}_{y_i}) - \max\left\{\left(\|\mathbf{A}_{y_i}^\top \mathbf{A}_{y_i}\|_F^2 \, \sigma_p^4 \frac{1}{C_1} \log(\frac{6n}{\delta})\right)^{1/2}, \|\mathbf{A}_{y_i}^\top \mathbf{A}_{y_i}\|_{\mathrm{op}} \sigma_p^2 \frac{1}{C_1} \log(\frac{6n}{\delta})\right\}.$$

In addition, by Lemmas B.5 and B.3, with probability at least $1 - \delta/3n^2$, there exists a constant $C_2 > 0$ such that

$$|\langle \boldsymbol{\xi}_{y_i}, \boldsymbol{\xi}_{y_j}\rangle| \leq \max\left\{\left(\|\mathbf{A}_{y_i}^\top \mathbf{A}_{y_j}\|_F^2 \, \sigma_p^4 \frac{1}{C_2} \log(\frac{6n^2}{\delta})\right)^{1/2}, \|\mathbf{A}_{y_i}^\top \mathbf{A}_{y_j}\|_{\mathrm{op}} \sigma_p^2 \frac{1}{C_2} \log(\frac{6n^2}{\delta})\right\}.$$

Applying $\sigma_p = \Theta(1)$ finishes the proof. $\qquad\square$

**Lemma B.14.** *Suppose that $d = \Omega(\log(mn/\delta))$ and $m = \Omega(\log(1/\delta))$. With probability at least $1-\delta$, for all $r \in [m], j \in [2], i \in [n]$,*

$$|\langle \mathbf{w}_{j,r}^{(0)}, \mathbf{u}_k\rangle| \leq \sqrt{2\log\left(\frac{4Km}{\delta}\right)} \|\mathbf{u}_k\|_2 \, \sigma_0, \tag{54}$$

$$|\langle \mathbf{w}_{j,r}^{(0)}, \boldsymbol{\xi}_i\rangle| \leq \mathcal{O}\left(\log\left(\frac{Km}{\delta}\right) \|\mathbf{A}_{y_i}\|_F \, \sigma_0\right). \tag{55}$$

*Proof.* We prove the first bound (54) with Hoeffding's inequality. For all $j \in [K], r \in [m], i \in [n]$, with probability $1 - \delta/(2Km)$,

$$|\langle \mathbf{w}_{j,r}^{(0)}, \mathbf{u}_k\rangle| \leq \sqrt{2\log\left(\frac{4Km}{\delta}\right)} \|\mathbf{u}_k\|_2 \, \sigma_0. \tag{56}$$

Then, we prove the second bound (55) leveraging Lemma B.5. For all $j \in [K], r \in [m], i \in [n]$, with probability $1 - \delta/(2Km)$,

$$\begin{aligned}
|\langle \mathbf{w}_{j,r}^{(0)}, \boldsymbol{\xi}_i\rangle| &\leq \mathcal{O}\left(\max\left\{\sqrt{\log\left(\frac{Km}{\delta}\right)} \|\mathbf{A}_{y_i}\|_F, \log\left(\frac{Km}{\delta}\right) \|\mathbf{A}_{y_i}\|_{\mathrm{op}}\right\} \sigma_0\right) \\
&\leq \mathcal{O}\left(\log\left(\frac{Km}{\delta}\right) \|\mathbf{A}_{y_i}\|_F \, \sigma_0\right).
\end{aligned} \tag{57}$$

$\qquad\square$

**Lemma B.15.** *Suppose $\delta > 0$ and $m \geq \Omega(\log(n/\delta))$. With probability at least $1 - \delta$, we have*

$$|\mathcal{S}_i^{(0)}| \geq 0.4m. \tag{58}$$

Lemma B.15 follows from Lemma B.6 and union bound.

**Lemma B.16.** *Suppose $\delta > 0$ and $n \geq \Omega(\log(m/\delta))$. For any neuron $\mathbf{w}_{j,r}^{(0)}, j \in [2], r \in [m]$, with probability at least $1 - \delta$, we have*

$$\sum_{i=1}^{n} \mathbb{I}(\langle \mathbf{w}_{j,r}^{(0)}, \boldsymbol{\xi}_i\rangle) \geq 0.4n. \tag{59}$$

Lemma B.16 follows from Lemma B.6 and union bound.

## C. Two-Stage Training Loss Analysis

In this section, we analyze the training loss. These results are based on the high probability conclusions in Appendix B. We divide the convergence of neural networks trained on each data into two stages. The training dynamics is at stage 1 at first and then enters stage 2 afterward. In Stage 1, we consider that the training data $(\mathbf{x}, y) \in \mathcal{S}$ satisfies $1 - \text{logit}_y(\mathbf{W}, \mathbf{x}) = \Theta(1)$.

### C.1. Network Gradient

The full-batch gradient on the neuron $\mathbf{w}_{k,r}$ at iteration $t$ is

$$
\nabla_{\mathbf{w}_{k,r}^{(t)}} \mathcal{L}(\mathbf{W}^{(t)}, \mathbf{x}, y) = -\frac{1}{mn} \sum_{(\mathbf{x},y)\in\mathcal{S}} \left[ \mathbb{I}(y = k)(1 - \text{logit}_k(\mathbf{W}^{(t)}, \mathbf{x})) \sum_{j=1}^{2} \sigma'\left(\langle \mathbf{w}_{k,r}^{(t)}, \mathbf{x}^{(j)}\rangle\right) \mathbf{x}^{(j)} \right]
$$

$$
+ \frac{1}{mn} \sum_{(\mathbf{x},y)\in\mathcal{S}} \left[ \mathbb{I}(y \neq k) \text{logit}_k(\mathbf{W}^{(t)}, \mathbf{x}) \sum_{j=1}^{2} \sigma'\left(\langle \mathbf{w}_{k,r}^{(t)}, \mathbf{x}^{(j)}\rangle\right) \mathbf{x}^{(j)} \right].
\tag{60}
$$

### C.2. Training Loss Bounds at Stage 1

First, we consider the training loss dynamics of stage 1 : For any data $(\mathbf{x}, y) \in \mathcal{S}$, we have $\mathcal{L}(\mathbf{W}^{(t')}, \mathbf{x}, y) = \Theta(1)$. First, we characterize the training loss.

$$
\begin{aligned}
&\mathcal{L}(\mathbf{W}^{(t+1)}, \mathbf{x}, y) - \mathcal{L}(\mathbf{W}^{(t)}, \mathbf{x}, y)\\
&= -\log\left(\text{prob}_y\left(\mathbf{W}^{(t+1)}, \mathbf{x}\right)\right) + \log\left(\text{prob}_y\left(\mathbf{W}^{(t)}, \mathbf{x}\right)\right)\\
&= \log\left(\frac{\text{prob}_y\left(\mathbf{W}^{(t)}, \mathbf{x}\right)}{\text{prob}_y\left(\mathbf{W}^{(t+1)}, \mathbf{x}\right)}\right)\\
&= \log\left(\frac{\exp\left(F_y^{(t)}(\mathbf{x})\right) / \left(\exp\left(F_y^{(t)}(\mathbf{x})\right) + \sum_{j\neq y}\exp\left(F_j^{(t)}(\mathbf{x})\right)\right)}{\exp\left(F_y^{(t+1)}(\mathbf{x})\right) / \left(\exp\left(F_y^{(t+1)}(\mathbf{x})\right) + \sum_{j\neq y}\exp\left(F_j^{(t+1)}(\mathbf{x})\right)\right)}\right)\\
&= \log\left(\frac{1 + \sum_{j\neq y}\exp\left(F_j^{(t)}(\mathbf{x}) - F_y^{(t)}(\mathbf{x})\right)\exp\left(\Delta_j^{(t)}(\mathbf{x}) - \Delta_y^{(t)}(\mathbf{x})\right)}{1 + \sum_{j\neq y}\exp\left(F_j^{(t)}(\mathbf{x}) - F_y^{(t)}(\mathbf{x})\right)}\right)\\
&= \log\left(1 + \frac{\sum_{j\neq y}\exp\left(F_j^{(t)}(\mathbf{x}) - F_y^{(t)}(\mathbf{x})\right)\left(\exp\left(\Delta_j^{(t)}(\mathbf{x}) - \Delta_y^{(t)}(\mathbf{x})\right) - 1\right)}{1 + \sum_{j\neq y}\exp\left(F_j^{(t)}(\mathbf{x}) - F_y^{(t)}(\mathbf{x})\right)}\right)\\
&\leq \log\left(1 + \left(1 - \text{prob}_y\left(\mathbf{W}^{(t)}, \mathbf{x}\right)\right)\max_{j\neq y}\left(\exp\left(\Delta_j^{(t)}(\mathbf{x}) - \Delta_y^{(t)}(\mathbf{x})\right) - 1\right)\right),
\end{aligned}
\tag{61}
$$

where $\Delta_y^{(t)}(\mathbf{x}) = F_y^{(t+1)}(\mathbf{x}) - F_y^{(t)}(\mathbf{x}), \Delta_j^{(t)}(\mathbf{x}) = F_j^{(t+1)}(\mathbf{x}) - F_j^{(t)}(\mathbf{x})$.

By Lemma B.7, we have

$$
\mathcal{L}\left(\mathbf{W}^{(t+1)}, \mathbf{x}, y\right) - \mathcal{L}\left(\mathbf{W}^{(t)}, \mathbf{x}, y\right) \leq \Theta(1) \cdot \max_{j\neq y}\left(\Delta_j^{(t)}(\mathbf{x}) - \Delta_y^{(t)}(\mathbf{x})\right).
\tag{62}
$$

To measure how training loss changes over iterations, we need to characterize the change of $\Delta_j^{(t)}(\mathbf{x}) - \Delta_y^{(t)}(\mathbf{x})$.

C.2.1. UPPER BOUND OF $\Delta_j^{(t)}(\mathbf{x}) - \Delta_y^{(t)}(\mathbf{x})$

We first rearrange $\Delta_j^{(t)}(\mathbf{x}) - \Delta_y^{(t)}(\mathbf{x})$ as follows.

$$
\begin{aligned}
&\Delta_j^{(t)}(\mathbf{x}) - \Delta_y^{(t)}(\mathbf{x}) \\
&= \frac{1}{m} \sum_{r=1}^m \sum_{i=1}^2 \left[ \sigma\left(\left\langle \mathbf{w}_{j,r}^{(t+1)}, \mathbf{x}^{(i)} \right\rangle\right) - \sigma\left(\left\langle \mathbf{w}_{j,r}^{(t)}, \mathbf{x}^{(j)} \right\rangle\right) \right] \\
&\quad - \frac{1}{m} \sum_{r=1}^m \sum_{i=1}^2 \left[ \sigma\left(\left\langle \mathbf{w}_{y,r}^{(t+1)}, \mathbf{x}^{(i)} \right\rangle\right) - \sigma\left(\left\langle \mathbf{w}_{y,r}^{(t)}, \mathbf{x}^{(i)} \right\rangle\right) \right] \\
&= \underbrace{\frac{1}{m} \sum_{r=1}^m \left[ \sigma\left(\left\langle \mathbf{w}_{j,r}^{(t+1)}, \mathbf{u}_y \right\rangle\right) - \sigma\left(\left\langle \mathbf{w}_{j,r}^{(t)}, \mathbf{u}_y \right\rangle\right) \right]}_{A} + \underbrace{\frac{1}{m} \sum_{r=1}^m \left[ \sigma\left(\left\langle \mathbf{w}_{j,r}^{(t+1)}, \boldsymbol{\xi} \right\rangle\right) - \sigma\left(\left\langle \mathbf{w}_{j,r}^{(t)}, \boldsymbol{\xi} \right\rangle\right) \right]}_{B} \\
&\quad \underbrace{- \frac{1}{m} \sum_{r=1}^m \left[ \sigma\left(\left\langle \mathbf{w}_{y,r}^{(t+1)}, \mathbf{u}_y \right\rangle\right) - \sigma\left(\left\langle \mathbf{w}_{y,r}^{(t)}, \mathbf{u}_y \right\rangle\right) \right]}_{C} \underbrace{- \frac{1}{m} \sum_{r=1}^m \left[ \sigma\left(\left\langle \mathbf{w}_{y,r}^{(t+1)}, \boldsymbol{\xi} \right\rangle\right) - \sigma\left(\left\langle \mathbf{w}_{y,r}^{(t)}, \boldsymbol{\xi} \right\rangle\right) \right]}_{D},
\end{aligned}
\tag{63}
$$

We then bound $A, B, C, D$ in Stage 1.

**Bound of $A$**

$$
\begin{aligned}
A &= \frac{1}{m} \sum_{r=1}^m \left[ \sigma\left( \left\langle \mathbf{w}_{j,r}^{(t)} - \frac{\eta}{mn} \sum_{(\mathbf{x}_i, y_i) \in \mathcal{S}_y} \sigma'(\langle \mathbf{w}_{j,r}^{(t)}, \mathbf{u}_y \rangle) \mathrm{logit}_j(\mathbf{W}^{(t)}, \mathbf{x}_i) \mathbf{u}_y, \mathbf{u}_y \right\rangle \right) \right] \\
&\quad - \frac{1}{m} \sum_{r=1}^m \left[ \sigma\left( \langle \mathbf{w}_{j,r}^{(t)}, \mathbf{u}_y \rangle \right) \right] \\
&\leq 0,
\end{aligned}
\tag{64}
$$

where the inequality is by the fact that $\langle \mathbf{u}_y, \mathbf{u}_y \rangle \geq 0$.

**Bound of $B$**

$$
\begin{aligned}
B &= \frac{1}{m} \sum_{r=1}^m \left[ \sigma\left( \left\langle \mathbf{w}_{j,r}^{(t)} - \frac{\eta}{mn} \sum_{(\mathbf{x}_i, y_i) \in \mathcal{S} \setminus \mathcal{S}_j} \sigma'\left( \langle \mathbf{w}_{j,r}^{(t)}, \boldsymbol{\xi}_i \rangle \right) \mathrm{logit}_j(\mathbf{W}^{(t)}, \mathbf{x}_i) \boldsymbol{\xi}_i \right. \right. \right. \\
&\quad \left. \left. \left. + \frac{\eta}{mn} \sum_{(\mathbf{x}_i, y_i) \in \mathcal{S}_j} \sigma'\left( \langle \mathbf{w}_{j,r}^{(t)}, \boldsymbol{\xi}_i \rangle \right) \left( 1 - \mathrm{logit}_j(\mathbf{W}^{(t)}, \mathbf{x}_i) \right) \boldsymbol{\xi}_i, \boldsymbol{\xi} \right\rangle \right) \right] - \frac{1}{m} \sum_{r=1}^m \left[ \sigma\left( \langle \mathbf{w}_{j,r}^{(t)}, \boldsymbol{\xi} \rangle \right) \right] \\
&\leq \frac{1}{m} \sum_{r=1}^m \left[ \sigma\left( \left\langle \mathbf{w}_{j,r}^{(t)}, \boldsymbol{\xi} \right\rangle - \frac{\eta}{mn} \sum_{(\mathbf{x}_i, y_i) \in \mathcal{S} \setminus \mathcal{S}_j} \sigma'\left( \langle \mathbf{w}_{j,r}^{(t)}, \boldsymbol{\xi}_i \rangle \right) \mathrm{logit}_j(\mathbf{W}^{(t)}, \mathbf{x}_i) \boldsymbol{\xi}_i \right. \right. \\
&\quad \left. \left. + \frac{\eta}{mn} \sum_{(\mathbf{x}_i, y_i) \in \mathcal{S}_j} \sigma'\left( \langle \mathbf{w}_{j,r}^{(t)}, \boldsymbol{\xi}_i \rangle \right) \left( 1 - \mathrm{logit}_j(\mathbf{W}^{(t)}, \mathbf{x}_i) \right) \boldsymbol{\xi}_i, \boldsymbol{\xi} \right\rangle \right) \right] - \frac{1}{m} \sum_{r=1}^m \left[ \sigma\left( \langle \mathbf{w}_{j,r}^{(t)}, \boldsymbol{\xi} \rangle \right) \right] \\
&\leq \frac{\eta}{mn} \sum_{(\mathbf{x}_i, y_i) \in \mathcal{S} \setminus \mathcal{S}_j} \sigma'\left( \langle \mathbf{w}_{j,r}^{(t)}, \boldsymbol{\xi}_i \rangle \right) \mathrm{logit}_j(\mathbf{W}^{(t)}, \mathbf{x}_i) |\langle \boldsymbol{\xi}_i, \boldsymbol{\xi} \rangle| \\
&\quad + \frac{\eta}{mn} \sum_{(\mathbf{x}_i, y_i) \in \mathcal{S}_j \setminus (\mathbf{x}, y)} \sigma'\left( \langle \mathbf{w}_{j,r}^{(t)}, \boldsymbol{\xi}_i \rangle \right) \left( 1 - \mathrm{logit}_j(\mathbf{W}^{(t)}, \mathbf{x}_i) \right) |\langle \boldsymbol{\xi}_i, \boldsymbol{\xi} \rangle|,
\end{aligned}
\tag{65}
$$

where the inequality is by the 1-Lipschitz property of the ReLU function.

**Bound of $C$**

$$
\begin{aligned}
C =& \frac{1}{m}\sum_{r=1}^{m}\left[\sigma\left(\left\langle \mathbf{w}_{y,r}^{(t)} + \frac{\eta}{mn}\sum_{(\mathbf{x}_i,y_i)\in\mathcal{S}_y}\sigma'(\langle\mathbf{w}_{y,r}^{(t)},\mathbf{u}_y\rangle)\left(1-\text{logit}_y(\mathbf{W}^{(t)},\mathbf{x}_i)\right)\mathbf{u}_y,\mathbf{u}_y\right\rangle\right)\right] \\
& - \frac{1}{m}\sum_{r=1}^{m}\left[\sigma\left(\langle\mathbf{w}_{y,r}^{(t)},\mathbf{u}_y\rangle\right)\right] \\
\geq & \frac{2\eta}{5mn}\sum_{(\mathbf{x},y)\in\mathcal{S}_y}\left(1-\text{logit}_y(\mathbf{W}^{(t)},\mathbf{x}_i)\right)\|\mathbf{u}_y\|_2^2,
\end{aligned}
\tag{66}
$$

where the inequality is by the fact that $\|\mathbf{u}_y\|_2^2 \geq 0$.

**Bound of $D$**

$$
\begin{aligned}
D =& \frac{1}{m}\sum_{r=1}^{m}\left[\sigma\left(\left\langle \mathbf{w}_{y,r}^{(t)} - \frac{\eta}{mn}\sum_{(\mathbf{x}_i,y_i)\in\mathcal{S}\backslash\mathcal{S}_y}\sigma'(\langle\mathbf{w}_{y,r}^{(t)},\boldsymbol{\xi}_i\rangle)\text{logit}_y(\mathbf{W}^{(t)},\mathbf{x}_i)\boldsymbol{\xi}_i\right.\right.\right. \\
& \left.\left.\left. + \frac{\eta}{mn}\sum_{(\mathbf{x}_i,y_i)\in\mathcal{S}_y}\sigma'\left(\langle\mathbf{w}_{y,r}^{(t)},\boldsymbol{\xi}_i\rangle\right)\left(1-\text{logit}_y(\mathbf{W}^{(t)},\mathbf{x}_i)\right)\boldsymbol{\xi}_i,\boldsymbol{\xi}\right\rangle\right)\right] \\
& - \frac{1}{m}\sum_{r=1}^{m}\left[\sigma\left(\langle\mathbf{w}_{y,r}^{(t)},\boldsymbol{\xi}\rangle\right)\right] \\
\geq & \frac{1}{m}\sum_{r=1}^{m}\left[\sigma\left(\left\langle\mathbf{w}_{y,r}^{(t)},\boldsymbol{\xi}\right\rangle - \frac{\eta}{mn}\sum_{(\mathbf{x}_i,y_i)\in\mathcal{S}\backslash\mathcal{S}_y}\sigma'(\langle\mathbf{w}_{y,r}^{(t)},\boldsymbol{\xi}_i\rangle)\text{logit}_y(\mathbf{W}^{(t)},\mathbf{x}_i)|\langle\boldsymbol{\xi}_i,\boldsymbol{\xi}\rangle|\right.\right. \\
& + \frac{\eta}{mn}\sigma'\left(\langle\mathbf{w}_{y,r}^{(t)},\boldsymbol{\xi}\rangle\right)\left(1-\text{logit}_y(\mathbf{W}^{(t)},\mathbf{x})\right)\|\boldsymbol{\xi}\|_2^2 \\
& \left.\left. - \frac{\eta}{mn}\sum_{(\mathbf{x}_i,y_i)\in\mathcal{S}_y\backslash(\mathbf{x},y)}\sigma'\left(\langle\mathbf{w}_{y,r}^{(t)},\boldsymbol{\xi}_i\rangle\right)\left(1-\text{logit}_y(\mathbf{W}^{(t)},\mathbf{x}_i)\right)|\langle\boldsymbol{\xi}_i,\boldsymbol{\xi}\rangle|\right)\right] \\
& - \frac{1}{m}\sum_{r=1}^{m}\left[\sigma\left(\langle\mathbf{w}_{y,r}^{(t)},\boldsymbol{\xi}_j\rangle\right)\right] \\
\geq & -\frac{\eta}{mn}\sum_{(\mathbf{x}_i,y_i)\in\mathcal{S}\backslash\mathcal{S}_y}\text{logit}_y(\mathbf{W}^{(t)},\mathbf{x}_i)|\langle\boldsymbol{\xi}_i,\boldsymbol{\xi}\rangle| + \frac{2\eta}{5mn}\left(1-\text{logit}_y(\mathbf{W}^{(t)},\mathbf{x})\right)\|\boldsymbol{\xi}\|_2^2 \\
& - \frac{\eta}{mn}\sum_{(\mathbf{x}_i,y_i)\in\mathcal{S}_y\backslash(\mathbf{x},y)}\left(1-\text{logit}_y(\mathbf{W}^{(t)},\mathbf{x}_i)\right)|\langle\boldsymbol{\xi}_i,\boldsymbol{\xi}\rangle| \\
\geq & 0,
\end{aligned}
\tag{67}
$$

where the last inequality is by that fact that the incremental term is larger than zero if a neuron is activated, by Lemmas 4.7, B.15, at least $0.4m$ neurons are activated, and $\sigma'(\cdot) \leq 1$.

Substituting (64), (65), (66), (67) into (63), by Lemma B.13 and Condition 3.1, we have

$$
\begin{aligned}
&\Delta_j^{(t)}(\mathbf{x}) - \Delta_y^{(t)}(\mathbf{x}) \\
&\leq -\frac{2\eta}{5mn} \sum_{(\mathbf{x},y)\in\mathcal{S}_y} \left(1 - \mathrm{logit}_y(\mathbf{W}^{(t)}, \mathbf{x}_i)\right) \|\mathbf{u}_y\|_2^2 \\
&\quad + \frac{2\eta}{mn} \sum_{(\mathbf{x}_i,y_i)\in\mathcal{S}\backslash\mathcal{S}_y} \mathrm{logit}_y(\mathbf{W}^{(t)}, \mathbf{x}_i) |\langle \boldsymbol{\xi}_i, \boldsymbol{\xi}\rangle| - \frac{2\eta}{5mn} \left(1 - \mathrm{logit}_y(\mathbf{W}^{(t)}, \mathbf{x})\right) \|\boldsymbol{\xi}\|_2^2 \\
&\quad + \frac{2\eta}{mn} \sum_{(\mathbf{x}_i,y_i)\in\mathcal{S}_y\backslash(\mathbf{x},y)} \left(1 - \mathrm{logit}_y(\mathbf{W}^{(t)}, \mathbf{x}_i)\right) |\langle \boldsymbol{\xi}_i, \boldsymbol{\xi}\rangle| \\
&\leq -\frac{2\eta}{5mn} \sum_{(\mathbf{x},y)\in\mathcal{S}_y} \left(1 - \mathrm{logit}_y(\mathbf{W}^{(t)}, \mathbf{x}_i)\right) \|\mathbf{u}_y\|_2^2 .
\end{aligned}
\tag{68}
$$

Combining (68) and (62), for any $(\mathbf{x}, y) \in \mathcal{S}$ we have

$$
\begin{aligned}
\mathcal{L}(\mathbf{W}^{(t+1)}, \mathbf{x}, y) &\leq \mathcal{L}(\mathbf{W}^{(t)}, \mathbf{x}, y) - \frac{2\eta}{5mn} \sum_{(\mathbf{x}_i,y_i)\in\mathcal{S}_y} \left(1 - \mathrm{logit}_y(\mathbf{W}^{(t)}, \mathbf{x}_i)\right) \|\mathbf{u}_y\|_2^2 \\
&\overset{(a)}{=} \left(1 - \Theta\left(\frac{\eta}{mn}|\mathcal{S}_y| \|\mathbf{u}_y\|_2^2\right)\right) \mathcal{L}(\mathbf{W}^{(t)}, \mathbf{x}, y),
\end{aligned}
\tag{69}
$$

where $(a)$ is obtained from Lemma 4.7. This finishes the training loss analysis in Stage 1.

### C.2.2. ANALYSIS OF STAGE 2

In stage 2, the loss of certain data is no longer $\Theta(1)$.

First, let $\mathbf{W}^*$ be

$$
\mathbf{w}_{j,r}^* = 2.5\log\left(\frac{4(K-1)}{\epsilon}\right) \frac{\mathbf{u}_j}{\|\mathbf{u}_j\|_2^2},
\tag{70}
$$

where $\epsilon > 0$ is a small constant. We have

$$
\left\|\mathbf{W}^{(t)} - \mathbf{W}^*\right\|_F^2 - \left\|\mathbf{W}^{(t+1)} - \mathbf{W}^*\right\|_F^2 = \underbrace{2\eta \left\langle \nabla\mathcal{L}_S(\mathbf{W}^{(t)}), \mathbf{W}^{(t)} - \mathbf{W}^*\right\rangle}_{A} - \underbrace{\eta^2 \left\|\nabla\mathcal{L}_S(\mathbf{W}^{(t)})\right\|_F^2}_{B}.
\tag{71}
$$

For the first term $A$, we have

$$
\begin{aligned}
A &= 2\eta \left\langle \nabla\mathcal{L}_S(\mathbf{W}^{(t)}), \mathbf{W}^{(t)} - \mathbf{W}^*\right\rangle \\
&= \frac{2\eta}{n} \sum_{(\mathbf{x},y)\in\mathcal{S}} \left\langle \nabla\mathcal{L}(\mathbf{W}^{(t)}, \mathbf{x}, y), \mathbf{W}^{(t)} - \mathbf{W}^*\right\rangle \\
&= \frac{2\eta}{n} \sum_{(\mathbf{x},y)\in\mathcal{S}} \sum_{k\in[K]} \left\langle \frac{\partial\mathcal{L}(\mathbf{W}^{(t)}, \mathbf{x}, y)}{\partial F_k} \nabla F_k(\mathbf{W}^{(t)}, \mathbf{x}, y), \mathbf{W}^{(t)} - \mathbf{W}^*\right\rangle \\
&= \frac{2\eta}{n} \sum_{(\mathbf{x},y)\in\mathcal{S}} \sum_{k\in[K]} \frac{\partial\mathcal{L}(\mathbf{W}^{(t)}, \mathbf{x}, y)}{\partial F_k} \left\langle \nabla F_k(\mathbf{W}^{(t)}, \mathbf{x}, y), \mathbf{W}^{(t)}\right\rangle \\
&\quad - \frac{2\eta}{n} \sum_{(\mathbf{x},y)\in\mathcal{S}} \sum_{k\in[K]} \frac{\partial\mathcal{L}(\mathbf{W}^{(t)}, \mathbf{x}, y)}{\partial F_k} \left\langle \nabla F_k(\mathbf{W}^{(t)}, \mathbf{x}, y), \mathbf{W}^*\right\rangle \\
&= \frac{2\eta}{n} \sum_{(\mathbf{x},y)\in\mathcal{S}} \sum_{k\in[K]} \frac{\partial\mathcal{L}(\mathbf{W}^{(t)}, \mathbf{x}, y)}{\partial F_k} \left( F_k(\mathbf{W}^{(t)}, \mathbf{x}, y) - \underbrace{\left\langle \nabla F_k(\mathbf{W}^{(t)}, \mathbf{x}, y), \mathbf{W}^*\right\rangle}_{C}\right),
\end{aligned}
\tag{72}
$$

where the last equality is based on the homogeneity of $F_k(\cdot)$ for any $k \in [K]$. Then, we need to bound $C$. First, for any $i, k \in [K]$, we have

$$\nabla_{\mathbf{w}_{k,r}^{(t)}} F_i(\mathbf{W}^{(t)}, \mathbf{x}, y) = \frac{1}{m} \left( \sigma' \left( \left\langle \mathbf{w}_{k,r}^{(t)}, \boldsymbol{\xi} \right\rangle \right) \boldsymbol{\xi} + \sigma' \left( \left\langle \mathbf{w}_{k,r}^{(t)}, \mathbf{u}_y \right\rangle \right) \mathbf{u}_y \right) \mathbb{I}(k = i). \tag{73}$$

By the definition of $\mathbf{W}^*$ in (70), we can get the components of $C$ as

$$\left\langle \nabla F_i(\mathbf{W}^{(t)}, \mathbf{x}, y), \mathbf{W}^* \right\rangle = \begin{cases} \frac{2.5 \log(4(K-1)/\epsilon)}{m} \sum_{r=1}^{m} \sigma' \left( \left\langle \mathbf{w}_{i,r}^{(t)}, \mathbf{u}_y \right\rangle \right) \|\mathbf{u}_y\|_2^2 & \text{if } i = y, \\ 0 & \text{if } i \neq y. \end{cases} \tag{74}$$

Next, we bound $B$.

$$\begin{aligned} B = &\eta^2 \left\| \nabla \mathcal{L}_S(\mathbf{W}^{(t)}) \right\|_F^2 \leq \eta^2 \left[ \frac{1}{n} \sum_{i=1}^{n} \left\| \sum_{k \in [K]} \frac{\partial \mathcal{L}(\mathbf{W}^{(t)}, \mathbf{x}, y)}{\partial F_k} \nabla F_k(\mathbf{W}^{(t)}, \mathbf{x}, y) \right\|_F \right]^2 \\ \leq &\eta^2 \left[ \frac{1}{n} \sum_{i=1}^{n} \sum_{k \in [K]} \left| \frac{\partial \mathcal{L}(\mathbf{W}^{(t)}, \mathbf{x}, y)}{\partial F_k} \right| \left\| \nabla F_k(\mathbf{W}^{(t)}, \mathbf{x}, y) \right\|_F \right]^2, \end{aligned} \tag{75}$$

where the first and the second inequalities are all based on triangle inequality. For any $k \in [K]$ and $(\mathbf{x}, y) \in \mathcal{S}$, we have

$$\begin{aligned} \left\| \nabla F_k(\mathbf{W}^{(t)}, \mathbf{x}, y) \right\|_F &= \left\| \sigma' \left( \left\langle \mathbf{w}_{k,r}^{(t)}, \mathbf{u}_y \right\rangle \right) \mathbf{u}_y + \sigma' \left( \left\langle \mathbf{w}_{k,r}^{(t)}, \boldsymbol{\xi} \right\rangle \right) \boldsymbol{\xi} \right\|_2 \\ &\leq \|\mathbf{u}_y\|_2 + \|\boldsymbol{\xi}\|_2 \\ &\leq \|\mathbf{u}_y\|_2 + \sqrt{\frac{3}{2} \text{Tr} \left( \mathbf{A}_y^\top \mathbf{A}_y \right)}, \end{aligned} \tag{76}$$

where the first inequality is by triangle inequality and the fact that $\sigma' \left( \left\langle \mathbf{w}_{k,r}^{(t)}, \mathbf{u}_y \right\rangle \right), \sigma' \left( \left\langle \mathbf{w}_{k,r}^{(t)}, \boldsymbol{\xi} \right\rangle \right) \leq 1$ and the second inequality is obtained from Lemma B.13.

Then, we can bound $B$ as

$$\begin{aligned} B \leq &\eta^2 \left[ \frac{1}{n} \sum_{i=1}^{n} \sum_{k \in [K]} \left| \frac{\partial \mathcal{L}(\mathbf{W}^{(t)}, \mathbf{x}, y)}{\partial F_k} \right| \left\| \nabla F_k(\mathbf{W}^{(t)}, \mathbf{x}, y) \right\|_F \right]^2 \\ \leq &\eta^2 \left( \max_{k \in [K]} \left( \|\mathbf{u}_k\|_2 + \sqrt{\frac{3}{2} \text{Tr} \left( \mathbf{A}_k^\top \mathbf{A}_k \right)} \right) \right)^2 \cdot 4 \left[ \frac{1}{n} \sum_{i=1}^{n} \left( 1 - \text{logit}_y \left( \mathbf{W}^{(t)}, \mathbf{x} \right) \right) \right]^2 \\ \leq &4\eta^2 \left( \max_{k \in [K]} \left( \|\mathbf{u}_k\|_2 + \sqrt{\frac{3}{2} \text{Tr} \left( \mathbf{A}_k^\top \mathbf{A}_k \right)} \right) \right)^2 \cdot \mathcal{L}_S(\mathbf{W}^{(t)}), \end{aligned} \tag{77}$$

where the first inequality is by (75), the second inequality is by (76) and the third inequality is by Jensen's inequality.

Combining them together, we have

$$\begin{aligned} A = &\frac{2\eta}{n} \sum_{(\mathbf{x}, y) \in \mathcal{S}} \sum_{k \in [K]} \frac{\partial \mathcal{L}(\mathbf{W}^{(t)}, \mathbf{x}, y)}{\partial F_i} \left( F_i(\mathbf{W}^{(t)}, \mathbf{x}, y) - \left\langle \nabla F_i(\mathbf{W}^{(t)}, \mathbf{x}, y), \mathbf{W}^* \right\rangle \right) \\ \geq &\frac{2\eta}{n} \sum_{(\mathbf{x}, y) \in \mathcal{S}} \left( \mathcal{L}(\mathbf{W}^{(t)}, \mathbf{x}, y) - \frac{\epsilon}{4} \right) \\ = &2\eta \left( \mathcal{L}_S(\mathbf{W}^{(t)}) - \frac{\epsilon}{4} \right), \end{aligned} \tag{78}$$

where the first inequality is based on the convex property (Lemma B.8) of the cross-entropy function and (74). Hence, we have

$$\left\|\mathbf{W}^{(t)} - \mathbf{W}^*\right\|_F^2 - \left\|\mathbf{W}^{(t+1)} - \mathbf{W}^*\right\|_F^2$$

$$\geq 2\eta \left(\mathcal{L}_S(\mathbf{W}^{(t)}) - \frac{\epsilon}{4}\right) - 4\eta^2 \left(\max_{k \in [K]} \left(\|\mathbf{u}_k\|_2 + \sqrt{\frac{3}{2}\mathrm{Tr}\left(\mathbf{A}_k^\top \mathbf{A}_k\right)}\right)\right)^2 \cdot \mathcal{L}_S(\mathbf{W}^{(t)}) \tag{79}$$

$$\geq \eta \mathcal{L}_S(\mathbf{W}^{(t)}) - \frac{\eta\epsilon}{2},$$

where the first inequality is by the bounds of $A$ and $B$ and the second inequality is obtained by the condition that $\eta \leq \left(4\left(\max_{k \in [K]}\left(\|\mathbf{u}_k\|_2 + \sqrt{1.5\mathrm{Tr}\left(\mathbf{A}_k^\top \mathbf{A}_k\right)}\right)\right)^2\right)^{-1}$.

Taking summation over all the iterations yields

$$\sum_{t=T_0}^{T} \eta \mathcal{L}_S(\mathbf{W}^{(t)}) \leq \left\|\mathbf{W}^{(T_0)} - \mathbf{W}^*\right\|_F^2 + \frac{(T - T_0 + 1)\eta\epsilon}{2}$$

$$\frac{1}{T - T_0 + 1}\sum_{t=T_0}^{T} \mathcal{L}_S(\mathbf{W}^{(t)}) \leq \frac{\left\|\mathbf{W}^{(T_0)} - \mathbf{W}^*\right\|_F^2}{\eta(T - T_0 + 1)} + \frac{\epsilon}{2}. \tag{80}$$

By the definition of $\mathbf{W}^*$, we have

$$\left\|\mathbf{W}^{(T_1)} - \mathbf{W}^*\right\|_F \leq \left\|\mathbf{W}^{(T_1)} - \mathbf{W}^{(0)}\right\|_F + \left\|\mathbf{0} - \mathbf{W}^{(0)}\right\|_F + \|\mathbf{0} - \mathbf{W}^*\|_F. \tag{81}$$

For $\left\|\mathbf{W}^{(T_1)} - \mathbf{W}^{(0)}\right\|_F$, we have

$$\left\|\mathbf{W}^{(T_1)} - \mathbf{W}^{(0)}\right\|_F \leq \mathcal{O}\left(T_1\frac{\eta}{m}\max_{i,j \in [K]}\{\|\mathbf{u}_i\|_2 + \mathrm{Tr}(\mathbf{A}_j^\top \mathbf{A}_j)\}\right) = \tilde{\mathcal{O}}\left(n\max_{j \in [K]}\{\mathrm{Tr}(\mathbf{A}_j^\top \mathbf{A}_j)\}\right). \tag{82}$$

Therefore, we have

$$\left\|\mathbf{W}^{(T_1)} - \mathbf{W}^*\right\|_F \leq \tilde{\mathcal{O}}\left(n\max_{j \in [K]}\{\mathrm{Tr}(\mathbf{A}_j^\top \mathbf{A}_j)\}\right) + \mathcal{O}(\sqrt{md}\sigma_0) + \mathcal{O}\left(\sqrt{mK}\log\left(\frac{4(K-1)}{\epsilon}\right)\right)$$

$$= \tilde{\mathcal{O}}\left(\max\{n\max_{j \in [K]}\{\mathrm{Tr}(\mathbf{A}_j^\top \mathbf{A}_j)\}, \sqrt{md}\sigma_0, \sqrt{mK}\}\right). \tag{83}$$

This finishes the training loss analysis in Stage 2.

## D. Test Loss Analysis

In this section, we analyze the test loss. Similar to the proof in Appendix C, the results are based on the high probability conclusions in Appendix B. In order to characterize the test loss, we first prove the following key lemmas.

### D.1. Key Lemmas for Test Loss Analysis

**Lemma D.1.** *Define*

$$\mathcal{S}_i^{(t)} = \{r \in [m] : \langle \mathbf{w}_{y_i,r}^{(t)}, \boldsymbol{\xi}_i \rangle > 0\}, \tag{84}$$

*for all* $(\mathbf{x}_i, y_i) \in \mathcal{S}$. *For any* $(\mathbf{x}_i, y_i), (\mathbf{x}_j, y_j) \in \mathcal{S}, t \in [T]$, *we have*

$$\frac{1 - \mathrm{logit}_{y_i}(\mathbf{W}^{(t)}, \mathbf{x}_i)}{1 - \mathrm{logit}_{y_j}(\mathbf{W}^{(t)}, \mathbf{x}_j)} \leq \kappa, \tag{85}$$

*for a constant* $\kappa > 1$ *and*

$$\mathcal{S}_i^{(t)} \subseteq \mathcal{S}_i^{(t+1)}. \tag{86}$$

*Proof.* We prove the first statement by induction. First, we show the conclusions hold at iteration 0. At iteration 0, with probability at least $1 - \delta$, for all $(\mathbf{x}, y) \in \mathcal{S}$, by Condition 3.1 and Lemma B.14, we have

$$0 \leq F_k(\mathbf{W}^{(0)}, \mathbf{x}) \leq C, \tag{87}$$

where $C > 0$ is a constant. Therefore, there exists a constant $\kappa > 0$ such that

$$\frac{1 - \mathrm{logit}_{y_i}(\mathbf{W}^{(0)}, \mathbf{x}_i)}{1 - \mathrm{logit}_{y_j}(\mathbf{W}^{(0)}, \mathbf{x}_j)} \leq \kappa. \tag{88}$$

Suppose there exists $\bar{t}$ such that the conditions hold for any $0 \leq t \leq \bar{t}$. We aim to prove that the conclusions also hold for $t = \bar{t} + 1$. We consider the following two cases.

Case 1: $\frac{1 - \mathrm{logit}_{y_i}(\mathbf{W}^{(t)}, \mathbf{x}_i)}{1 - \mathrm{logit}_{y_j}(\mathbf{W}^{(t)}, \mathbf{x}_j)} < 0.9\kappa$. First, we have

$$\frac{1 - \mathrm{logit}_{y_i}\left(\mathbf{W}^{(t+1)}, \mathbf{x}_i\right)}{1 - \mathrm{logit}_{y_i}(\mathbf{W}^{(t)}, \mathbf{x}_i)} = \frac{\sum_{k \neq y_i} \exp(F_k^{(t)}(\mathbf{x}_i)) \exp(\Delta_k^{(t)}(\mathbf{x}_i))}{\sum_{k \neq y_i} \exp(F_k^{(t)}(\mathbf{x}_i))} \frac{\sum_{k \in [K]} \exp\left(F_k^{(t)}(\mathbf{x}_i)\right)}{\sum_{k \in [K]} \exp\left(F_k^{(t)}(\mathbf{x}_i)\right) \exp\left(\Delta_k^{(t)}(\mathbf{x}_i)\right)}, \tag{89}$$

indicating that

$$\frac{1 - \mathrm{logit}_{y_i}\left(\mathbf{W}^{(t+1)}, \mathbf{x}_i\right)}{1 - \mathrm{logit}_{y_i}(\mathbf{W}^{(t)}, \mathbf{x}_i)} \leq \frac{\max_{k \in [K]} \exp\left(\Delta_k^{(t)}(\mathbf{x}_i)\right)}{\min_{k \in [K]} \exp\left(\Delta_k^{(t)}(\mathbf{x}_i)\right)}. \tag{90}$$

Moreover, we have

$$|\Delta_k^{(t)}(\mathbf{x}_i)| \leq \eta \max_{k \in [K]} \left\{\|\mathbf{u}_k\|_2^2 + \|\boldsymbol{\xi}\|_2^2\right\} \leq \eta \max_{k \in [K]} \left\{\|\mathbf{u}_k\|_2^2 + \frac{3}{2}\mathrm{Tr}\left(\mathbf{A}_k^\top \mathbf{A}_k\right)\right\} \leq 0.02, \tag{91}$$

where the inequalities are by Lemma B.13 and Condition 3.1. Then, we have

$$\begin{aligned}
&\frac{1 - \mathrm{logit}_{y_i}\left(\mathbf{W}^{(t+1)}, \mathbf{x}_i\right)}{1 - \mathrm{logit}_{y_j}\left(\mathbf{W}^{(t+1)}, \mathbf{x}_j\right)} \\
&= \frac{1 - \mathrm{logit}_{y_i}\left(\mathbf{W}^{(t+1)}, \mathbf{x}_i\right)}{1 - \mathrm{logit}_{y_i}\left(\mathbf{W}^{(t)}, \mathbf{x}_i\right)} \frac{1 - \mathrm{logit}_{y_j}\left(\mathbf{W}^{(t)}, \mathbf{x}_j\right)}{1 - \mathrm{logit}_{y_j}\left(\mathbf{W}^{(t+1)}, \mathbf{x}_j\right)} \frac{1 - \mathrm{logit}_{y_i}\left(\mathbf{W}^{(t)}, \mathbf{x}_i\right)}{1 - \mathrm{logit}_{y_j}\left(\mathbf{W}^{(t)}, \mathbf{x}_j\right)} \\
&\leq \frac{1 - \mathrm{logit}_{y_i}\left(\mathbf{W}^{(t)}, \mathbf{x}_i\right)}{1 - \mathrm{logit}_{y_j}\left(\mathbf{W}^{(t)}, \mathbf{x}_j\right)} \cdot \exp(0.09) \\
&\leq \kappa.
\end{aligned} \tag{92}$$

Case 2: $\frac{1-\text{logit}_{y_i}\left(\mathbf{W}^{(t)}, \mathbf{x}_i\right)}{1-\text{logit}_{y_j}\left(\mathbf{W}^{(t)}, \mathbf{x}_j\right)} > 0.9\kappa > 1$. We have

$$
\begin{aligned}
&\frac{1 - \text{logit}_{y_i}\left(\mathbf{W}^{(t+1)}, \mathbf{x}_i\right)}{1 - \text{logit}_{y_j}\left(\mathbf{W}^{(t+1)}, \mathbf{x}_j\right)} \\
&\leq \frac{\sum_{k\neq y_j} \exp(F_k^{(t)}(\mathbf{x}_j)) + \exp\left(F_{y_j}^{(t)}(\mathbf{x}_j)\right)\exp\left(\Delta_{y_j}^{(t)}(\mathbf{x}_j) - \min_{k\neq y_j}\Delta_k^{(t)}(\mathbf{x}_j)\right)}{\sum_{k\neq y_i} \exp(F_k^{(t)}(\mathbf{x}_i)) + \exp\left(F_{y_i}^{(t)}(\mathbf{x}_i)\right)\exp\left(\Delta_{y_i}^{(t)}(\mathbf{x}_i) - \max_{k\neq y_i}\Delta_k^{(t)}(\mathbf{x}_i)\right)} \\
&\quad \frac{\sum_{k\neq y_i}\exp(F_k^{(t)}(\mathbf{x}_i))}{\sum_{k\neq y_j}\exp(F_k^{(t)}(\mathbf{x}_j))} \\
&= \frac{1 + \left(\frac{1}{1-\text{logit}_{y_j}(\mathbf{W}^{(t)},\mathbf{x}_j)} - 1\right)\exp\left(\Delta_{y_j}^{(t)}(\mathbf{x}_j) - \min_{k\neq y_j}\Delta_k^{(t)}(\mathbf{x}_j)\right)}{1 + \left(\frac{1}{1-\text{logit}_{y_i}(\mathbf{W}^{(t)},\mathbf{x}_i)} - 1\right)\exp\left(\Delta_{y_i}^{(t)}(\mathbf{x}_i) - \max_{k\neq y_i}\Delta_k^{(t)}(\mathbf{x}_i)\right)} \\
&\leq \max\{1, \kappa\exp(\Delta_{y_j}^{(t)}(\mathbf{x}_j) - \min_{k\neq y_j}\Delta_k^{(t)}(\mathbf{x}_j) - \Delta_{y_i}^{(t)}(\mathbf{x}_i) + \max_{k\neq y_i}\Delta_k^{(t)}(\mathbf{x}_i))\}
\end{aligned}
\tag{93}
$$

Denote $l_1 = \arg\min_{k\neq y_j}\Delta_k^{(t)}(\mathbf{x}_j)$ and $l_2 = \arg\max_{k\neq y_i}\Delta_k^{(t)}(\mathbf{x}_i)$. We have

$$
\begin{aligned}
&\Delta_{l_2}^{(t)}(\mathbf{x}_i) - \Delta_{y_i}^{(t)}(\mathbf{x}_i) - \Delta_{l_1}^{(t)}(\mathbf{x}_j) + \Delta_{y_j}^{(t)}(\mathbf{x}_j) \\
&\leq \frac{\eta}{n}\sum_{k\neq i}(1 - \text{logit}_{y_k}(\mathbf{W}^{(t)}, \mathbf{x}_k))|\langle \boldsymbol{\xi}_i, \boldsymbol{\xi}_k\rangle| - \frac{2\eta}{5n}(1 - \text{logit}_{y_i}(\mathbf{W}^{(t)}, \mathbf{x}_i))\|\boldsymbol{\xi}_i\|_2^2 \\
&\quad + \frac{\eta}{n}\sum_{k\neq j}(1 - \text{logit}_{y_k}(\mathbf{W}^{(t)}, \mathbf{x}_k))|\langle \boldsymbol{\xi}_j, \boldsymbol{\xi}_k\rangle| + \frac{\eta}{n}(1 - \text{logit}_{y_j}(\mathbf{W}^{(t)}, \mathbf{x}_j))\|\boldsymbol{\xi}_j\|_2^2 \\
&\quad - \frac{2\eta}{5n}(1 - \text{logit}_{y_i}(\mathbf{W}^{(t)}, \mathbf{x}_i))\|\mathbf{u}_i\|_2^2 + \frac{\eta}{n}(1 - \text{logit}_{y_j}(\mathbf{W}^{(t)}, \mathbf{x}_j))\|\mathbf{u}_j\|_2^2 \\
&\leq 0,
\end{aligned}
\tag{94}
$$

by letting $\kappa > \Theta(\max\{\text{Tr}(\mathbf{A}_{y_j}^\top \mathbf{A}_{y_j})/\text{Tr}(\mathbf{A}_{y_i}^\top \mathbf{A}_{y_i}), \|\mathbf{u}_j\|_2^2 / \|\mathbf{u}_i\|_2^2\})$. Then the last inequality is by $\frac{1-\text{logit}_{y_i}\left(\mathbf{W}^{(t)},\mathbf{x}_i\right)}{1-\text{logit}_{y_j}\left(\mathbf{W}^{(t)},\mathbf{x}_j\right)} > 0.9\kappa$ and Lemma B.13. Therefore, we have

$$
\frac{1 - \text{logit}_{y_i}\left(\mathbf{W}^{(t+1)}, \mathbf{x}_i\right)}{1 - \text{logit}_{y_j}\left(\mathbf{W}^{(t+1)}, \mathbf{x}_j\right)} \leq \kappa.
\tag{95}
$$

This completes the proof of the induction.

Next, we prove the second statement. For a data sample $(\mathbf{x}_i, y_i) \in \mathcal{S}$ and a neuron in $\mathcal{S}_i^{(t)}$, we have

$$
\begin{aligned}
\langle \mathbf{w}_{y_i,r}^{(t+1)}, \boldsymbol{\xi}_i\rangle &= \langle \mathbf{w}_{y_i,r}^{(t)}, \boldsymbol{\xi}_i\rangle + \frac{\eta}{mn}(1 - \text{logit}_{y_i}(\mathbf{x}_i))\sigma'(\langle \mathbf{w}_{y_i,r}^{(t)}, \boldsymbol{\xi}_i\rangle)\|\boldsymbol{\xi}_i\|_2^2 \\
&\quad - \frac{\eta}{mn}\sum_{i'\neq i}\text{logit}_{y_i}(\mathbf{x}_i)\sigma'(\langle \mathbf{w}_{y_i,r}^{(t)}, \boldsymbol{\xi}_{i'}\rangle)\langle \boldsymbol{\xi}_{i'}, \boldsymbol{\xi}_i\rangle \\
&\geq \langle \mathbf{w}_{y_i,r}^{(t)}, \boldsymbol{\xi}_i\rangle,
\end{aligned}
\tag{96}
$$

where the inequality is by Lemma B.13, Condition 3.1 and (88). Thus, we have $\mathcal{S}_i^{(t)} \subseteq \mathcal{S}_i^{(t+1)}$. This completes the proof. $\square$

**Lemma D.2.** *Under Condition 3.1, for any $j \in [K], l \in [K] \backslash \{j\}, (\mathbf{x}_q, y_q) \in \mathcal{S}_j, (\mathbf{x}_a, y_a) \in \mathcal{S}_l,$ and $r \in \mathcal{S}_q^{(0)}$,*

$$
\sum_{t'=0}^{t-1} (1 - \text{logit}_{y_q}(\mathbf{W}^{(t')}, \mathbf{x}_q)) \sigma'(\langle \mathbf{w}_{j,r}^{(t')}, \boldsymbol{\xi}_q \rangle)
$$
$$
= \Omega \left( \frac{1}{n} \sum_{t'=0}^{t-1} (\text{logit}_j(\mathbf{W}^{(t')}, \mathbf{x}_a)) \sigma'(\langle \mathbf{w}_{j,r}^{(t')}, \boldsymbol{\xi}_a \rangle) \frac{\text{Tr}(\mathbf{A}_l^\top \mathbf{A}_l)}{\max_{l_1, l_2 \in [K]} \left\| \mathbf{A}_{l_1}^\top \mathbf{A}_{l_2} \right\|_F^{1/2} \log(3n^2/\delta)^{1/2}} \right). \tag{97}
$$

*Proof.* By the update rule of gradient descent, we have

$$
\langle \mathbf{w}_{j,r}^{(t+1)} - \mathbf{w}_{j,r}^{(t)}, \boldsymbol{\xi}_a \rangle \leq \frac{\eta}{mn} \sum_{(\mathbf{x}_i, y_i) \in \mathcal{S} \backslash (\mathbf{x}_a, y_a)} \left( 1 - \text{logit}_j(\mathbf{W}^{(t)}, \boldsymbol{\xi}_i) \right) \sigma'(\langle \mathbf{w}_{j,r}^{(t)}, \boldsymbol{\xi}_i \rangle) |\langle \boldsymbol{\xi}_i, \boldsymbol{\xi}_a \rangle|
$$
$$
+ \frac{\eta}{mn} (-\text{logit}_y(\mathbf{W}^{(t)}, \mathbf{x})) \sigma'(\langle \mathbf{w}_{j,r}^{(t)}, \boldsymbol{\xi}_a \rangle) \| \boldsymbol{\xi}_a \|_2^2 . \tag{98}
$$

By Lemma B.13, we have

$$
\langle \mathbf{w}_{j,r}^{(t+1)} - \mathbf{w}_{j,r}^{(0)}, \boldsymbol{\xi}_a \rangle + \frac{\eta}{mn} \sum_{t'=0}^{t-1} (\text{logit}_j(\mathbf{W}^{(t')}, \mathbf{x}_a)) \sigma'(\langle \mathbf{w}_{j,r}^{(t')}, \boldsymbol{\xi}_a \rangle) \| \boldsymbol{\xi}_a \|_2^2
$$
$$
\leq \frac{\eta}{mn} \sum_{(\mathbf{x}, y) \in \mathcal{S} \backslash (\mathbf{x}_a, y_a)} \sum_{t'=0}^{t} (1 - \text{logit}_y(\mathbf{W}^{(t')}, \mathbf{x})) \underbrace{\max_{l, y \in [K]} \left\| \mathbf{A}_l^\top \mathbf{A}_y \right\|_F^{1/2} \log(\frac{3n^2}{\delta})^{1/2}}_{E_1} . \tag{99}
$$

Additionally, by the nature of ReLU activation function, the magnitude of $\langle \mathbf{w}_{j,r}^{(t+1)} - \mathbf{w}_{j,r}^{(0)}, \boldsymbol{\xi}_a \rangle$ satisfies

$$
\langle \mathbf{w}_{j,r}^{(t+1)} - \mathbf{w}_{j,r}^{(0)}, \boldsymbol{\xi}_a \rangle \geq -\frac{\eta}{mn} \sum_{(\mathbf{x}, y) \in \mathcal{S} \backslash (\mathbf{x}_a, y_a)} \sum_{t'=0}^{t} (1 - \text{logit}_y(\mathbf{W}^{(t')}, \mathbf{x})) E_1 - \frac{\eta}{mn} \| \boldsymbol{\xi}_a \|_2^2 . \tag{100}
$$

As the learnign rate $\eta$ is small (by Condition 3.1), combining (99) and (100), for any $(\mathbf{x}, y) \sim \mathcal{D}$, we have

$$
\frac{\eta}{mn} \sum_{t'=0}^{t-1} (\text{logit}_j(\mathbf{W}^{(t')}, \mathbf{x}_a)) \sigma'(\langle \mathbf{w}_{j,r}^{(t')}, \boldsymbol{\xi}_a \rangle) \| \boldsymbol{\xi}_a \|_2^2
$$
$$
= \mathcal{O} \left( \frac{\eta}{mn} \sum_{(\mathbf{x}, y) \in \mathcal{S} \backslash (\mathbf{x}_a, y_a)} \sum_{t'=0}^{t} (1 - \text{logit}_y(\mathbf{W}^{(t')}, \mathbf{x})) E_1 \right), \tag{101}
$$

By Lemma D.1, for $r \in \mathcal{S}_q^{(0)}$, we have

$$
\sum_{t'=0}^{t-1} (1 - \text{logit}_{y_q}(\mathbf{W}^{(t')}, \mathbf{x}_q)) \sigma'(\langle \mathbf{w}_{j,r}^{(t')}, \boldsymbol{\xi}_q \rangle)
$$
$$
= \Omega \left( \frac{1}{n} \sum_{t'=0}^{t-1} (\text{logit}_j(\mathbf{W}^{(t')}, \mathbf{x}_a)) \sigma'(\langle \mathbf{w}_{j,r}^{(t')}, \boldsymbol{\xi}_a \rangle) \frac{\| \boldsymbol{\xi}_a \|_2^2}{E_1} \right). \tag{102}
$$

By Lemma B.13, we have

$$
\frac{\| \boldsymbol{\xi}_a \|_2^2}{E_1} = \Omega \left( \frac{\text{Tr}(\mathbf{A}_l^\top \mathbf{A}_l)}{\max_{l_1, l_2 \in [K]} \left\| \mathbf{A}_{l_1}^\top \mathbf{A}_{l_2} \right\|_F^{1/2} \log(3n^2/\delta)^{1/2}} \right). \tag{103}
$$

Combining (102) and (103) yields the conclusion. This completes the proof. □

**Lemma D.3.** *Under Condition 3.1, for any $j \in [K]$, we have*

$$\left\| \left( \mathbf{W}_j^{(T)} \right)^{\top} \mathbf{A}_j \right\|_2 \leq \mathcal{O}(\sqrt{m} \log(T)). \tag{104}$$

*Proof.* We bound it with two phases. The first phase $t \in [0, \tilde{T}_1]$ is before the loss of data $(\mathbf{x}, y) \in \mathcal{S}$ satisfying

$$\mathrm{logit}_y(\mathbf{x}) \leq c_5, \tag{105}$$

where $c_5$ is a constant. Then, the loss satisfies

$$
\begin{aligned}
&(1 - c_5) \exp \left( \sigma \left( \left\langle \mathbf{w}_{y,r}^{(\tilde{T}_1)}, \boldsymbol{\xi} \right\rangle \right) \right) \\
\leq &(1 - c_5) \exp \left( \sigma \left( \left\langle \mathbf{w}_{y,r}^{(\tilde{T}_1)}, \mathbf{u}_y \right\rangle \right) + \sigma \left( \left\langle \mathbf{w}_{y,r}^{(\tilde{T}_1)}, \boldsymbol{\xi} \right\rangle \right) \right) \\
\leq &c_5 \sum_{j \neq y} \exp \left( \sigma \left( \left\langle \mathbf{w}_{j,r}^{(\tilde{T}_1)}, \mathbf{u}_j \right\rangle \right) + \sigma \left( \left\langle \mathbf{w}_{j,r}^{(\tilde{T}_1)}, \boldsymbol{\xi} \right\rangle \right) \right) \\
\leq &c_5 K \exp \left( \sqrt{2 \log \left( \frac{4Km}{\delta} \right)} \|\mathbf{u}_k\|_2 \, \sigma_0 + \mathcal{O} \left( \log \left( \frac{Km}{\delta} \right) \|\mathbf{A}_y\|_F \, \sigma_0 \right) + \frac{\eta}{mn} \|\boldsymbol{\xi}\|_2^2 \right) \\
\leq &1.1 c_5 K \exp \left( \frac{3\eta}{2mn} \mathrm{Tr} \left( \mathbf{A}_y^{\top} \mathbf{A}_y \right) \right),
\end{aligned}
\tag{106}
$$

where the first inequality is by the fact that $\sigma \left( \left\langle \mathbf{w}_{y,r}^{(\tilde{T}_1)}, \mathbf{u}_y \right\rangle \right) \geq 0$, the second inequality is by the definition of softmax function and (105), the third inequality is by Lemma B.14 and the fourth inequality is by Lemma B.13 and Condition 3.1. Letting $c_5$ be $T/(1.1K + T)$ yields

$$\left\langle \mathbf{w}_{y,r}^{(\tilde{T}_1)}, \boldsymbol{\xi} \right\rangle \leq \log \left( \frac{c_5}{1 - c_5} K \right) + \frac{3\eta}{2mn} \mathrm{Tr} \left( \mathbf{A}_y^{\top} \mathbf{A}_y \right) \leq \mathcal{O} \left( \log(T) \right). \tag{107}$$

Moreover, for $T > \tilde{T}_1$, we have

$$
\begin{aligned}
\langle \mathbf{w}_{y,r}^{(T)}, \boldsymbol{\xi} \rangle =& \langle \mathbf{w}_{y,r}^{(\tilde{T}_1)}, \boldsymbol{\xi} \rangle + \langle \mathbf{w}_{y,r}^{(T)} - \mathbf{w}_{y,r}^{(\tilde{T}_1)}, \boldsymbol{\xi} \rangle \\
\leq & \mathcal{O} \left( \log(T) \right) + 2 \sum_{t=\tilde{T}_1+1}^{T} \frac{\eta}{mn} (1 - \mathrm{logit}_y(\mathbf{W}^{(t)}, \mathbf{x})) \|\boldsymbol{\xi}\|_2^2 \\
\leq & \mathcal{O} \left( \log(T) \right) + \frac{3\eta T}{mn} \frac{1}{T} \mathrm{Tr} \left( \mathbf{A}_y^{\top} \mathbf{A}_y \right) \\
\leq & \mathcal{O}(\log(T)),
\end{aligned}
\tag{108}
$$

where the first inequality is by (107), the second inequality is by Lemma B.13 and condition of the second phase that $\mathrm{logit}_y(\mathbf{x}) > c_5$. With probability of $1 - \delta$, for $n$ randomly sampled data $(\mathbf{x}', y) \sim \mathcal{D}_y$, we have

$$
\begin{aligned}
\langle \mathbf{W}_{y,r}^{(T)}, \boldsymbol{\xi}' \rangle \leq & \mathcal{O} \left( \frac{\eta}{mm} \sum_{i \in \mathcal{S}} \sum_{t=0}^{T-1} (1 - \mathrm{logit}(\mathbf{W}^{(t)}, \mathbf{x})) |\langle \boldsymbol{\xi}_i, \boldsymbol{\xi}' \rangle| \right) \\
\leq & \mathcal{O} \left( \frac{\eta}{mn} \sum_{t=0}^{T-1} (1 - \mathrm{logit}(\mathbf{W}^{(t)}, \mathbf{x})) \|\boldsymbol{\xi}\|_2^2 \right) = \Theta(\langle \mathbf{w}_{y,r}^{(T)}, \boldsymbol{\xi} \rangle) = \left( \mathbf{w}_{y,r}^{(T)} \right)^{\top} \mathbf{A}_y \boldsymbol{\zeta},
\end{aligned}
\tag{109}
$$

With probability $1 - \delta$, at least one sample $(\mathbf{x}', y)$ satisfies $\langle \mathbf{w}_{y,r}^{(T)}, \boldsymbol{\xi}' \rangle = \Theta \left( \left\| \left( \mathbf{w}_{y,r}^{(T)} \right)^{\top} \mathbf{A}_y \right\|_2 \right)$ by the property of $\mathcal{D}_\zeta$. Therefore, we have

$$\langle \mathbf{w}_{y,r}^{(T)}, \boldsymbol{\xi} \rangle = \Omega \left( \left\| \left( \mathbf{w}_{y,r}^{(T)} \right)^{\top} \mathbf{A}_y \right\|_2 \right). \tag{110}$$

Combining (108) and (110) completes the proof. $\qquad \square$

**Lemma D.4.** *Under condition 3.1, for a random vector $\boldsymbol{\xi}$ generated from $\mathbf{A}_j\boldsymbol{\zeta}$, $\boldsymbol{\zeta} \sim \mathcal{D}_\zeta$ for any $j \in [K]$, with probability at least $1 - \delta$, we have*

$$\sum_{r=1}^{m} \mathbb{I}\left(\langle \mathbf{w}_{j,r}^{(T)}, \boldsymbol{\xi}\rangle\right) \geq 0.1m. \tag{111}$$

*Proof.* First, we can concatenate the neuron weights for class $j \in [K]$ as

$$\mathbf{W}_j^{(T)}\mathbf{A}_j = \begin{bmatrix} \left(\mathbf{w}_{j,1}^{(T)}\right)^\top \\ \vdots \\ \left(\mathbf{w}_{j,m}^{(T)}\right)^\top \end{bmatrix} \mathbf{A}_j = \underbrace{\begin{bmatrix} \left(\mathbf{w}_{j,1}^{(0)}\right)^\top \\ \vdots \\ \left(\mathbf{w}_{j,m}^{(0)}\right)^\top \end{bmatrix}}_{\mathbf{D}_1} \mathbf{A}_j + \underbrace{\begin{bmatrix} \beta_{j,1,1} & \cdots & \beta_{j,1,n} \\ \vdots & \vdots & \vdots \\ \beta_{j,m,1} & \cdots & \beta_{j,m,n} \end{bmatrix} \begin{bmatrix} \boldsymbol{\xi}_1^\top \\ \vdots \\ \boldsymbol{\xi}_n^\top \end{bmatrix}}_{\mathbf{D}_2} \mathbf{A}_j, \tag{112}$$

where $\beta_{j,r,i} := \sum_{t'=0}^{T-1} \sigma'(\langle \mathbf{w}_{j,r}^{(t')}, \boldsymbol{\xi}_i\rangle)\text{logit}(\mathbf{W}^{(t')}, \mathbf{x}_i)$. The rank of matrix $\mathbf{D}_2\mathbf{A}_j$ satisfies

$$\text{rank}(\mathbf{D}_2\mathbf{A}_j) \leq \min\{m, n, d, \text{rank}(\mathbf{A}_j)\} = n. \tag{113}$$

In addition, as the matrix $\mathbf{D}_1$ is a Gaussian random matrix, it has full rank almost surely. We have

$$\text{rank}(\mathbf{D}_1\mathbf{A}_j) = \min\{m, d, \text{rank}(\mathbf{A}_j)\}, \tag{114}$$

almost surely. Then by Condition 3.1, the rank of $\mathbf{W}_j^{(T)}\mathbf{A}_j$ satisfies

$$\text{rank}(\mathbf{W}_j^{(T)}\mathbf{A}_j) \geq \min\{m, d, \text{rank}(\mathbf{A}_j)\} - n \geq 0.9m. \tag{115}$$

In addition, by Lemma B.1 and Condition 3.1, the singular value of $\mathbf{W}_j^{(T)}\mathbf{A}_j$ satisfies

$$\lambda_{\min\{m,d,\text{rank}(\mathbf{A}_j)\}-n}(\mathbf{W}_j^{(T)}\mathbf{A}_j) \geq 0.1\sqrt{m}\sigma_0. \tag{116}$$

Moreover, by Lemma D.3, we have

$$\lambda_1(\mathbf{W}_j^{(T)}\mathbf{A}_j) \leq \left\|\mathbf{W}_j^{(T)}\mathbf{A}_j\right\|_F \leq \mathcal{O}(\sqrt{m}\log(T)). \tag{117}$$

Therefore, according to Condition 3.1, we have

$$m \geq \Omega\left(\frac{\log(n/\delta)\log(T)^2}{n\sigma_0^2}\right). \tag{118}$$

By Lemma B.12 and Condition 3.1, with probability at least $1 - \delta$, we have

$$\sum_{r=1}^{m} \mathbb{I}\left(\langle \mathbf{w}_{j,r}^{(T)}, \boldsymbol{\xi}\rangle\right) \geq 0.1m. \tag{119}$$

This completes the proof. $\square$

### D.1.1. PROOF OF STATEMENT 2(A) IN THEOREM 3.4

In this part, we prove Statement 2(a) in Theorem 3.4. For data samples following $(\mathbf{x}, y) \sim \mathcal{D}$, the test loss satisfies

$$\begin{aligned} L_\mathcal{D}(\mathbf{W}^{(t)}) &= \mathbb{P}\left[\arg\max_k F_k(\mathbf{W}^{(T)}, \mathbf{x}, y) \neq y\right] \\ &\leq \sum_{j\neq y} \mathbb{P}\left[F_y(\mathbf{W}^{(T)}, \mathbf{x}, y) \leq F_j(\mathbf{W}^{(T)}, \mathbf{x}, y)\right] \\ &= \sum_{j\neq y} \mathbb{P}\left[\sum_{r=1}^{m} \sigma(\langle \mathbf{w}_{y,r}^{(T)}, \mathbf{u}_y\rangle) + \sigma(\langle \mathbf{w}_{y,r}^{(T)}, \boldsymbol{\xi}\rangle) \leq \sum_{r=1}^{m} \sigma(\langle \mathbf{w}_{j,r}^{(T)}, \mathbf{u}_y\rangle) + \sigma(\langle \mathbf{w}_{j,r}^{(T)}, \boldsymbol{\xi}\rangle)\right]. \end{aligned} \tag{120}$$

For the features, we bound the loss for any $(\mathbf{x}, y) \sim \mathcal{D}$ through

$$\mathcal{L}(\mathbf{W}^{(T)}, \mathbf{x}, y) \le \sum_{j \ne y} \mathbb{P} \left[ \sum_{r=1}^{m} \sigma(\langle \mathbf{w}_{y,r}^{(T)}, \mathbf{u}_y \rangle) \le \sum_{r=1}^{m} \sigma(\langle \mathbf{w}_{j,r}^{(T)}, \mathbf{u}_y \rangle) + \sigma(\langle \mathbf{w}_{j,r}^{(T)}, \boldsymbol{\xi} \rangle) \right]. \tag{121}$$

Then, we can bound the model outputs as

$$
\begin{aligned}
&\sum_{r=1}^{m} \sigma \left( \left\langle \mathbf{w}_{y,r}^{(T)}, \mathbf{u}_y \right\rangle \right) \\
&= \sum_{r=1}^{m} \sigma \left( \left\langle \mathbf{w}_{y,r}^{(0)} + \frac{\eta}{n} \sum_{t'=1}^{T} \sum_{(\mathbf{x}_l, y_l) \in \mathcal{S}_y} (1 - \mathrm{logit}_y(\mathbf{W}^{(t')}, \mathbf{x}_l)) \sigma'(\langle \mathbf{w}_{y,r}^{(t')}, \mathbf{u}_y \rangle) \mathbf{u}_y, \mathbf{u}_y \right\rangle \right) \\
&\ge \frac{3m}{10} \frac{\eta}{n} \sum_{t'=1}^{T} \sum_{(\mathbf{x}_l, y_l) \in \mathcal{S}_y} (1 - \mathrm{logit}_y(\mathbf{W}^{(t')}, \mathbf{x}_l)) \|\mathbf{u}_y\|_2^2,
\end{aligned} \tag{122}
$$

and

$$\sum_{r=1}^{m} \sigma(\langle \mathbf{w}_{j,r}^{(T)}, \boldsymbol{\xi} \rangle) \le \sum_{r=1}^{m} |\langle \mathbf{w}_{j,r}^{(T)}, \boldsymbol{\xi} \rangle| = \sum_{r=1}^{m} \left\| \mathbf{A}_k^{\top} \mathbf{w}_{j,r}^{(T)} \right\|_2 |\zeta'|, \tag{123}$$

where $\zeta'$ is a sub-Gaussian variable. Suppose that $\delta > 0$ and $\left\| \mathbf{A}_y^{\top} \mathbf{A}_j \right\|_F / \left\| \mathbf{A}_y^{\top} \mathbf{A}_j \right\|_{\mathrm{op}} \ge \Theta(\sqrt{\log(K^2/\delta)})$. For the term $\left\| \mathbf{A}_y^{\top} \mathbf{w}_{j,r}^{(T)} \right\|_2$, with probability at least $1 - \delta$, we have

$$
\begin{aligned}
&\left\| \mathbf{A}_y^{\top} \mathbf{w}_{j,r}^{(T)} \right\|_2 \\
&= \left\| \mathbf{A}_y^{\top} \left( \mathbf{w}_{j,r}^{(0)} + \frac{\eta}{n} \sum_{t'=0}^{T-1} \sum_{(\mathbf{x},y) \in \mathcal{S}_j} (1 - \mathrm{logit}_j(\mathbf{W}^{(t')}, \mathbf{x})) \sigma'(\langle \mathbf{w}_{j,r}^{(t')}, \boldsymbol{\xi} \rangle) \boldsymbol{\xi} \right. \right. \\
&\qquad \left. \left. - \frac{\eta}{n} \sum_{(\mathbf{x},y) \in \mathcal{S} \backslash \mathcal{S}_j} \mathrm{logit}_j(\mathbf{W}^{(t')}, \mathbf{x}) \sigma'(\langle \mathbf{w}_{j,r}^{(t')}, \boldsymbol{\xi} \rangle) \boldsymbol{\xi} \right) \right\|_2 \\
&\le \left\| \mathbf{A}_y^{\top} \mathbf{A}_j \left( \frac{\eta}{n} \sum_{t'=0}^{T-1} \sum_{(\mathbf{x},y) \in \mathcal{S}_j} (1 - \mathrm{logit}_j(\mathbf{W}^{(t')}, \mathbf{x})) \sigma'(\langle \mathbf{w}_{j,r}^{(t')}, \boldsymbol{\xi} \rangle) \boldsymbol{\zeta} \right) \right\|_2 \\
&\quad + \left\| \sum_{k \ne j} \mathbf{A}_y^{\top} \mathbf{A}_k \left( \frac{\eta}{n} \sum_{(\mathbf{x},y) \in \mathcal{S}_k} \mathrm{logit}_j(\mathbf{W}^{(t')}, \mathbf{x}) \sigma'(\langle \mathbf{w}_{j,r}^{(t')}, \boldsymbol{\xi} \rangle) \boldsymbol{\zeta} \right) \right\|_2 \\
&\overset{(a)}{=} \Theta \left( \left\| \mathbf{A}_y^{\top} \left( \frac{\eta}{n} \sum_{t'=0}^{T-1} \sum_{(\mathbf{x},y) \in \mathcal{S}_j} (1 - \mathrm{logit}_j(\mathbf{W}^{(t')}, \mathbf{x})) \sigma'(\langle \mathbf{w}_{j,r}^{(t')}, \boldsymbol{\xi} \rangle) \boldsymbol{\xi} \right) \right\|_2 \right) \\
&\overset{(b)}{=} \Theta \left( \frac{\eta}{n} \left\| \mathbf{A}_y^{\top} \mathbf{A}_j \right\|_F \cdot \sqrt{\sum_{(\mathbf{x},y) \in \mathcal{S}_j} \left( \sum_{t'=0}^{T-1} (1 - \mathrm{logit}_j(\mathbf{W}^{(t')}, \mathbf{x})) \right)^2} \right),
\end{aligned} \tag{124}
$$

where the inequality is by Lemma D.2 and Condition 3.1, $(a)$ is based on Lemma D.2, and $(b)$ is based on the concentration of random vectors (Theorem 6.2.6 in (Vershynin, 2018)). Substituting (122) and (123) into (121), for any $(\mathbf{x}, y) \sim \mathcal{D}$ we

have

$$\mathcal{L}(\mathbf{W}^{(T)}, \mathbf{x}, y)$$

$$\leq \sum_{j \neq y} \mathbb{P}\left[ \frac{m}{5} \frac{\eta}{n} \sum_{t'=0}^{T-1} \sum_{(\mathbf{x},y) \in \mathcal{S}_y} (1 - \mathrm{logit}_y(\mathbf{W}^{(t')}, \mathbf{x})) \|\mathbf{u}_y\|_2^2 \leq \sum_{r=1}^{m} |\langle \mathbf{w}_{j,r}^{(T)}, \boldsymbol{\xi} \rangle| + |\langle \mathbf{w}_{j,r}^{(T)}, \mathbf{u}_y \rangle| \right]$$

$$= \sum_{j \neq y} \mathbb{P}\left[ \frac{m}{5} \frac{\eta}{n} \sum_{t'=0}^{T-1} \sum_{(\mathbf{x},y) \in \mathcal{S}_y} (1 - \mathrm{logit}_y(\mathbf{W}^{(t')}, \mathbf{x})) \|\mathbf{u}_y\|_2^2 \leq \sum_{r=1}^{m} \left\|\mathbf{A}_y^\top \mathbf{w}_{j,r}^{(T)}\right\|_2 |\zeta'| + |\langle \mathbf{w}_{j,r}^{(0)}, \mathbf{u}_y \rangle| \right]$$

$$\leq \sum_{j \neq y} \mathbb{P}\left[ \sum_{t'=0}^{T-1} \sum_{(\mathbf{x},y) \in \mathcal{S}_y} (1 - \mathrm{logit}_y(\mathbf{W}^{(t')}, \mathbf{x})) \|\mathbf{u}_y\|_2^2 \right. \tag{125}$$

$$\leq c \left\|\mathbf{A}_y^\top \mathbf{A}_j\right\|_F \cdot \sqrt{\sum_{(\mathbf{x},y) \in \mathcal{S}_j} \left( \sum_{t'=0}^{T-1} (1 - \mathrm{logit}_j(\mathbf{W}^{(t')}, \mathbf{x})) \right)^2} |\zeta'| + \mathcal{O}\left( \frac{n}{\eta} \sqrt{\log\left( \frac{4Km}{\delta} \right)} \|\mathbf{u}_y\|_2 \sigma_0 \right) \right]$$

$$\leq \sum_{j \neq y} \exp\left( -c_1 \cdot \frac{|\mathcal{S}_y|^2 \|\mathbf{u}_y\|_2^4}{|\mathcal{S}_j| \left\|\mathbf{A}_y^\top \mathbf{A}_j\right\|_F^2} \right),$$

where $c, c_1 > 0$ are some constants and the last inequality is obtained from Hoeffding's inequality.

### D.1.2. PROOF OF STATEMENT 2(B) IN THEOREM 3.4

Furthermore, for long-tailed data distribution, by Lemma D.2 and union bound, we have

$$\mathcal{L}^{0-1}(\mathbf{W}^{(t)}, \mathbf{x}, y) \leq \sum_{j \neq y} \mathbb{P}\left[ \sum_{r=1}^{m} \sigma\left( \left\langle \mathbf{w}_{y,r}^{(t)}, \boldsymbol{\xi} \right\rangle \right) \leq \sum_{r=1}^{m} \sigma\left( \left\langle \mathbf{w}_{j,r}^{(T)}, \mathbf{u}_y \right\rangle \right) + \sigma\left( \left\langle \mathbf{w}_{j,r}^{(t)}, \boldsymbol{\xi} \right\rangle \right) \right]. \tag{126}$$

Suppose $\delta > 0$, by Condition 3.1, w have $\left\|\mathbf{A}_y^\top \mathbf{A}_y\right\|_F / \left\|\mathbf{A}_y^\top \mathbf{A}_y\right\|_{\mathrm{op}} \geq \Theta(\sqrt{\log(K^2/\delta)})$, for any $y \in [K]$. With probability at least $1 - \delta$, for all $y \in [K]$, we have

$$\left\|\mathbf{A}_y^\top \mathbf{w}_{y,r}^{(T)}\right\|_2$$

$$= \left\|\mathbf{A}_y^\top \left( \mathbf{w}_{y,r}^{(0)} + \frac{\eta}{n} \sum_{t'=0}^{T-1} \sum_{(\mathbf{x},y) \in \mathcal{S}_y} (1 - \mathrm{logit}_y(\mathbf{W}^{(t')}, \mathbf{x})) \sigma'(\langle \mathbf{w}_{y,r}^{(t')}, \boldsymbol{\xi} \rangle) \boldsymbol{\xi} \right. \right.$$

$$\left. \left. - \frac{\eta}{n} \sum_{t'=0}^{T-1} \sum_{(\mathbf{x},y) \in \mathcal{S} \setminus \mathcal{S}_y} \mathrm{logit}_y(\mathbf{W}^{(t')}, \mathbf{x}) \sigma'(\langle \mathbf{w}_{y,r}^{(t')}, \boldsymbol{\xi} \rangle) \boldsymbol{\xi} \right) \right\|_2$$

$$\geq \left\|\mathbf{A}_y^\top \mathbf{A}_y \left( \frac{\eta}{n} \sum_{t'=0}^{T-1} \sum_{(\mathbf{x},y) \in \mathcal{S}_y} (1 - \mathrm{logit}_y(\mathbf{W}^{(t')}, \mathbf{x})) \sigma'(\langle \mathbf{w}_{y,r}^{(t')}, \boldsymbol{\xi} \rangle) \boldsymbol{\zeta} \right) \right\|_2 \tag{127}$$

$$- \left\| \sum_{j \neq y} \mathbf{A}_y^\top \mathbf{A}_j \left( \frac{\eta}{n} \sum_{t'=0}^{T-1} \sum_{(\mathbf{x},y) \in \mathcal{S}_j} \mathrm{logit}_y(\mathbf{W}^{(t')}, \mathbf{x}) \sigma'(\langle \mathbf{w}_{y,r}^{(t')}, \boldsymbol{\xi} \rangle) \boldsymbol{\zeta} \right) \right\|_2$$

$$\overset{(a)}{=} \Omega\left( \frac{\eta}{n} \left\|\mathbf{A}_y^\top \mathbf{A}_y\right\|_F \cdot \sqrt{\sum_{(\mathbf{x},y) \in \mathcal{S}_y} \left( \sum_{t'=0}^{T-1} (1 - \mathrm{logit}_y(\mathbf{W}^{(t')}, \mathbf{x})) \right)^2} \right),$$

where $(a)$ is obtained by the condition $\left\|\mathbf{A}_y^\top \mathbf{A}_y\right\|_F = \Omega\left( \left\|\mathbf{A}_y^\top \mathbf{A}_j\right\|_F \right)$ for all $j, y \in [K]$ and $j \neq y$, $K = \Theta(1)$, and (85).

Similarly, we also have $\left\|\mathbf{A}_y^\top \mathbf{w}_{y,r}^{(T)}\right\|_2 \leq \mathcal{O}\left( \eta/n \left\|\mathbf{A}_y^\top \mathbf{A}_y\right\|_F \cdot \sqrt{\sum_{(\mathbf{x},y) \in \mathcal{S}_y} \left( \sum_{t'=0}^{T-1} (1 - \mathrm{logit}_y(\mathbf{W}^{(t')}, \mathbf{x})) \right)^2} \right)$. Then,

we have

$$\left\| \mathbf{A}_y^\top \mathbf{w}_{y,r}^{(T)} \right\|_2 = \Theta \left( \frac{\eta}{n} \left\| \mathbf{A}_y^\top \mathbf{A}_y \right\|_F \cdot \sqrt{\sum_{(\mathbf{x},y) \in \mathcal{S}_y} \left( \sum_{t'=0}^{T-1} (1 - \text{logit}_y(\mathbf{W}^{(t')}, \mathbf{x})) \right)^2} \right). \tag{128}$$

Similar to the proof of (124), we also have

$$\left\| \mathbf{A}_y^\top \mathbf{w}_{j,r}^{(T)} \right\|_2 = \Theta \left( \frac{\eta}{n} \left\| \mathbf{A}_y^\top \mathbf{A}_j \right\|_F \cdot \sqrt{\sum_{(\mathbf{x},y) \in \mathcal{S}_j} \left( \sum_{t'=0}^{T-1} (1 - \text{logit}_j(\mathbf{W}^{(t')}, \mathbf{x})) \right)^2} \right), \tag{129}$$

with probability at least $1 - \delta$. Moreover, we have

$$\left\| \mathbf{A}_y^\top \mathbf{w}_{j,r}^{(T)} \right\|_2 = \Theta \left( \frac{\eta}{n} \left\| \mathbf{A}_y^\top \mathbf{A}_j \right\|_F \cdot \sqrt{\sum_{(\mathbf{x},y) \in \mathcal{S}_j} \left( \sum_{t'=0}^{T-1} (1 - \text{logit}_j(\mathbf{W}^{(t')}, \mathbf{x})) \right)^2} \right). \tag{130}$$

Denote $\bar{\mathcal{S}}_j(\boldsymbol{\xi}) = \{r \in [m] : \langle \mathbf{w}_{j,r}^{(T)}, \boldsymbol{\xi} \rangle > 0\}$. We have

$$\left\| \sum_{r \in \bar{\mathcal{S}}_y(\boldsymbol{\xi})} \mathbf{A}_y^\top \mathbf{w}_{y,r}^{(T)} \right\|_2 = \Theta \left( \frac{\eta m}{n} \left\| \mathbf{A}_y^\top \mathbf{A}_j \right\|_F \cdot \sqrt{\sum_{(\mathbf{x},y) \in \mathcal{S}_j} \left( \sum_{t'=0}^{T-1} (1 - \text{logit}_j(\mathbf{W}^{(t')}, \mathbf{x})) \right)^2} \right), \tag{131}$$

because Lemma 4.4 holds.

We have

$$
\begin{aligned}
\mathcal{L}^{0-1}(\mathbf{W}^{(T)}, \mathbf{x}, y) &\leq \sum_{j \neq y} \mathbb{P} \left[ \frac{1}{m} \sum_{r=1}^m \sigma(\langle \mathbf{w}_{y,r}^{(T)}, \boldsymbol{\xi} \rangle) \leq \frac{1}{m} \sum_{r=1}^m \sigma(\langle \mathbf{w}_{j,r}^{(T)}, \mathbf{u}_y \rangle) + \sigma(\langle \mathbf{w}_{j,r}^{(T)}, \boldsymbol{\xi} \rangle) \right] \\
&= \sum_{j \neq y} \mathbb{P} \left[ | \sum_{r \in \bar{\mathcal{S}}_y(\boldsymbol{\xi})} \langle \mathbf{w}_{y,r}^{(T)}, \boldsymbol{\xi} \rangle | \leq \sum_{r=1}^m |\langle \mathbf{w}_{j,r}^{(T)}, \boldsymbol{\xi} \rangle| + |\langle \mathbf{w}_{j,r}^{(T)}, \mathbf{u}_y \rangle| \right] \\
&\leq \sum_{j \neq y} \mathbb{P} \left[ L \cdot \left\| \sum_{r \in \bar{\mathcal{S}}_y(\boldsymbol{\xi})} \mathbf{A}_y^\top \mathbf{w}_{y,r}^{(T)} \right\|_2 - \sum_{r=1}^m \left\| \mathbf{A}_y^\top \mathbf{w}_{j,r}^{(T)} \right\|_2 \leq \sum_{r=1}^m |\langle \mathbf{w}_{j,r}^{(T)}, \boldsymbol{\xi} \rangle| - \sum_{r=1}^m \mathbb{E}|\langle \mathbf{w}_{j,r}^{(T)}, \boldsymbol{\xi} \rangle| + |\langle \mathbf{w}_{j,r}^{(0)}, \mathbf{u}_y \rangle| \right] \\
&\leq \sum_{j \neq y} \exp \left( -c' \cdot \frac{\left( L \cdot \left\| \sum_{r \in \bar{\mathcal{S}}_y(\boldsymbol{\xi})} \mathbf{A}_y^\top \mathbf{w}_{y,r}^{(T)} \right\|_2 - \sum_{r=1}^m \left\| \mathbf{A}_y^\top \mathbf{w}_{j,r}^{(T)} \right\|_2 - \mathcal{O}\left( \sqrt{\log(4Km/\delta)} \|\mathbf{u}_y\|_2 \sigma_0 \right) \right)^2}{\sum_{r=1}^m \left\| \mathbf{A}_y^\top \mathbf{w}_{j,r}^{(T)} \right\|_2^2} \right) \\
&\leq \sum_{j \neq y} \exp \left( -c_2 L^2 \cdot \frac{|\mathcal{S}_y| \left\| \mathbf{A}_y^\top \mathbf{A}_y \right\|_F^2}{|\mathcal{S}_j| \left\| \mathbf{A}_y^\top \mathbf{A}_j \right\|_F^2} \right),
\end{aligned}
\tag{132}
$$

where $c' > 0$ is a constant, the first inequality is by the condition for long-tailed data that $|\langle \mathbf{w}_{y,r}^{(T)}, \boldsymbol{\xi} \rangle| \geq c_5 \left\| \mathbf{A}_y^\top \mathbf{w}_{y,r}^{(T)} \right\|_2$, the third inequality is by Hoeffding's inequality and the last inequality is by (128) and (129).

D.1.3. TEST LOSS LOWER BOUND (STATEMENT 3 IN THEOREM 3.4)

For any $(\mathbf{x}, y) \sim \mathcal{D}$, we have

$$
\begin{aligned}
\mathcal{L}^{0-1}(\mathbf{W}^{(T)}, \mathbf{x}, y) &\geq \max_{j \neq y} \mathbb{P} \left[ \sum_{r=1}^{m} \sigma(\langle \mathbf{w}_{y,r}^{(T)}, \mathbf{u}_y \rangle) + \sigma(\langle \mathbf{w}_{y,r}^{(T)}, \boldsymbol{\xi} \rangle) \leq \sum_{r=1}^{m} \sigma(\langle \mathbf{w}_{j,r}^{(T)}, \mathbf{u}_y \rangle) + \sigma(\langle \mathbf{w}_{j,r}^{(T)}, \boldsymbol{\xi} \rangle) \right] \\
&= \max_{j \neq y} \mathbb{P} \left[ \sum_{r=1}^{m} \sigma(\langle \mathbf{w}_{y,r}^{(T)}, \mathbf{u}_y \rangle) + \sigma(\langle \mathbf{w}_{y,r}^{(T)}, \boldsymbol{\xi} \rangle) \leq \sum_{r=1}^{m} \sigma(\langle \mathbf{w}_{j,r}^{(T)}, \mathbf{u}_y \rangle) + \sigma(\langle \mathbf{w}_{j,r}^{(T)}, \boldsymbol{\xi} \rangle) \right] \\
&\overset{(a)}{\geq} \max_{j \neq y} \mathbb{P} \Bigg[ \frac{2\eta m}{5n} \sum_{t'=0}^{T-1} \sum_{(\mathbf{x},y) \in \mathcal{S}_y} (1 - \mathrm{logit}_y(\mathbf{W}^{(t')}, \mathbf{x})) \|\mathbf{u}_y\|_2^2 \\
&\quad + \underbrace{\Theta \left( \frac{\eta m}{n} \|\mathbf{A}_y^\top \mathbf{A}_y\|_F \cdot \sqrt{\sum_{(\mathbf{x},y) \in \mathcal{S}_y} \left( \sum_{t'=0}^{T-1} (1 - \mathrm{logit}_y(\mathbf{W}^{(t')}, \mathbf{x})) \right)^2} \right) |\zeta'|}_{B_1} \\
&\quad \leq \underbrace{\Theta \left( \frac{\eta m}{n} \|\mathbf{A}_y^\top \mathbf{A}_j\|_F \cdot \sqrt{\sum_{(\mathbf{x},y) \in \mathcal{S}_j} \left( \sum_{t'=0}^{T-1} (1 - \mathrm{logit}_j(\mathbf{W}^{(t')}, \mathbf{x})) \right)^2} \right) |\zeta''|}_{B_2} \Bigg],
\end{aligned}
\tag{133}
$$

where $(a)$ is based on (127) and (124). $\zeta'$ and $\zeta''$ are sub-Gaussain variables with variance 1. For any $(\mathbf{x}, y) \sim \mathcal{D}$, when there exists $j \in [K]$ and a constant $C_3$ satisfying $n \|\mathbf{u}_k\|_2^2 \leq C_3 \|\mathbf{A}_k^\top \mathbf{A}_j\|_F$ and $\|\mathbf{A}_k^\top \mathbf{A}_k\|_F \leq C_3 \|\mathbf{A}_k^\top \mathbf{A}_j\|_F$, by Lemma B.10, we have

$$
\mathcal{L}^{0-1}(\mathbf{W}^{(T)}, \mathbf{x}, y) \geq \max_{j \neq y} \mathbb{P} \left[ \frac{2\eta m}{5n} \sum_{(\mathbf{x},y) \in \mathcal{S}_y} (1 - \mathrm{logit}_y(\mathbf{W}^{(t')}, \mathbf{x})) \|\mathbf{u}_y\|_2^2 \leq B_2 |\zeta''| - B_1 |\zeta'| \right]
\tag{134}
$$

Choosing a constant $c_5 = \Theta(\sigma_p)$, we have $|\zeta'| \leq c_5$ with probability at least $1 - \exp(\Omega(c_5^2/\sigma_p^2))$, resulting in

$$
\begin{aligned}
\mathcal{L}^{0-1}(\mathbf{W}^{(T)}, \mathbf{x}, y) &\geq \max_{j \neq y} \mathbb{P} \left[ \Theta \left( \sum_{t'=0}^{T-1} \sqrt{|\mathcal{S}_y|} \|\mathbf{u}_y\|_2^2 \right) \leq \Theta(\|\mathbf{A}_y^\top \mathbf{A}_j\|_F)|\zeta''| - \Theta(\|\mathbf{A}_y^\top \mathbf{A}_y\|_F)c_5 \right] \left( 1 - \exp(\Omega(c_5^2/\sigma_p^2)) \right) \\
&= \Omega(1).
\end{aligned}
\tag{135}
$$

# E. Computation of Squared Frobenius Norm for Real-World Datasets

To estimate the quantity $\mathbf{A}_i$ for real-world datasets, we decompose the intra-class data covariance matrix. Recall that in our data model, the covariance matrix of a data sample $(\mathbf{x}, y)$ within class $i$ is

$$
\boldsymbol{\Sigma}_i = \mathbf{A}_i \mathbf{A}_i^\top.
\tag{136}
$$

To estimate the $\mathbf{A}_i$ for real-world datasets, we first compute the sample covariance matrix:

$$
\tilde{\boldsymbol{\Sigma}}_i = \frac{1}{|\mathcal{S}_i| - 1} \sum_{(\mathbf{x},y) \in \mathcal{S}_i} (\mathbf{x} - \bar{\mathbf{x}})(\mathbf{x} - \bar{\mathbf{x}})^\top,
\tag{137}
$$

where $|\mathcal{S}_i|$ is the size of class $i$ in the training dataset and $\bar{\mathbf{x}}$ is the sample mean.

Next, we perform an eigendecomposition of the sample covariance matrices $\tilde{\boldsymbol{\Sigma}}_i = \mathbf{Q}_i \boldsymbol{\Lambda}_i \mathbf{Q}_i^\top$, where $\mathbf{Q}_i$ contains the eigenvectors and $\boldsymbol{\Lambda}_i$ is the diagonal matrix of eigenvalues.

We then estimate $\mathbf{A}_i$ as $\tilde{\mathbf{A}}_i = \mathbf{Q}_i \boldsymbol{\Lambda}_i^{1/2}$. Using the estimated $\tilde{\mathbf{A}}_i$ for each class, we can estimate $\|\mathbf{A}_i^\top \mathbf{A}_j\|_F$.

