# OpenReview forum: "Rethinking Benign Overfitting in Two-Layer Neural Networks"
_ICML.cc/2025/Conference — ICML 2025 poster_

### Official Review · Reviewer_5giJ · 2025-03-07

**Overall Recommendation:** 3

**Summary:**

This paper analyzes the training dynamics of a two-layer CNN which can lead to benign overfitting. By analyzing a new data model with feature-specific noise, they claim that neural networks can learn "implicit features" that improve the accuracy when training on long-tailed data.

**Claims And Evidence:**

The theoretical results in section 3 on the test loss for the two-layer CNN and the experiments in section 5 support the paper's main claim about benign overfitting in the proposed data model.

One claim that I find slightly misleading is that this paper shows that memorizing data noise, which was previously deemed harmful, can help classification accuracy. The "data noise" in the model presented in this behavior is class-dependent and can essentially be interpreted as containing class-specific information that transfers over to the dataset. This is different from prior works, in which the "data noise" class-independent noise. From this point of view, it is not surprising that memorizing data noise can help in the setting presented in this paper while being harmful in other works, and I am not sure to what extent calling the two different types of noise both "data noise" is accurate.

**Essential References Not Discussed:**

None that I am aware of.

**Experimental Designs Or Analyses:**

The design and analysis of the experiments on the synthetic dataset in Section 5.1 make sense and support the theoretical results from the previous section.

I do not quite understand the noise correlation verification in section 5.2, in particular how $||A_i^{\top}A_j||_F$ is calculated for real-world datasets such as MNIST and CIFAR-10 when these datasets do not exactly follow the data model presented in section 3 for which the quantities are defined.

**Methods And Evaluation Criteria:**

The analysis of the test loss in Section 3 makes sense for the problem statement. The construction of the synthetic dataset in Section 5.1 mirrors the theoretical setting, and the benchmark datasets for noise correlation (MNIST, CIFAR-10) are commonly used.

**Other Comments Or Suggestions:**

See above.

**Other Strengths And Weaknesses:**

This paper provides an interesting new perspective on benign overfitting by considering a data model with a new form of class-dependent feature noise. The paper is somewhat easy to follow, although some of the assumptions (e.g. Condition 1) are a bit opaque at first glance and could use more explanation.

**Questions For Authors:**

My main questions are

1. Making comparisons about the effect of data noise when the "data noise" is fundamentally different in this work.

2. How $||A_i^{\top}A_j||_F$  is calculated for real-world datasets.

**Relation To Broader Scientific Literature:**

This paper mainly belongs to a line of work theoretically analyzing benign overfitting in neural networks. The main difference in this work is the new data model, from which the authors derive new results about the effect of memorizing said noise.

The authors also connect some of their results to the spurious correlation literature by showing that their results imply that class imbalance can hurt group accuracy.

One interesting connection which the authors do not mention is to adversarial robustness. Specifically, the feature hypothesis "that adversarial examples can be directly attributed to the presence of non-robust features: features (derived from patterns in the data distribution) that are highly predictive, yet brittle and (thus) incomprehensible to humans" [1].  Indeed, if what the authors call data noise or implicit features can be recast as non-robust features, then there could be some connection there. I would be curious about the authors' thoughts on the same.

[1] Ilyas, A., Santurkar, S., Tsipras, D., Engstrom, L., Tran, B., & Madry, A. (2019). Adversarial examples are not bugs, they are features.

**Theoretical Claims:**

Proofs were checked at a high level, details were not carefully checked.

---

> ### Author Rebuttal · Authors · 2025-04-01
>
> We sincerely appreciate the reviewer for recognizing our contributions and for the constructive feedbacks.
> Below are our point-by-point responses:
>
> **Clarification on "Data Noise" Definition**:
> We agree with your point and will revise the term 'noise' to 'data-specific information' and clarify its class-dependent nature.
> We will also clarify this distinction throughout the paper to avoid any potential confusion.
>
> **Relationship to Adversarial Robustness**:
> We agree with your assessment that the implicit features discussed in our paper could be interpreted as non-robust features.
> As demonstrated in paper [1], standard training learns non-robust features from data-specific information, while adversarial training purifies these features to enhance model robustness.
>
> In this context, our theoretical framework has the potential to extend results in [1].
> Specifically, while adversarial training removes non-robust features, our paper posits that this process negatively impacts plain accuracy, particularly in long-tailed data. This corollary is empirically supported by the findings in [2], further supporting our theory.
>
> **Comparison with Other Noise Models**:
> We acknowledge the extensive research on the effect of label and data noise on benign overfitting. To clarify the distinctions between our model and prior work, we provide a comparison with existing studies on benign overfitting, focusing on two-layer non-linear neural networks incorporating data noise.
>
> Studies [3-7] (a representative selection due to space constraints) focus on the 'explicit feature + class-independent Gaussian data noise' model (similar to the feature-noise model developed by [8]).
> While these studies explore various training settings, they consistently show that excessive data-specific noise leads to overfitting.
> However, they fail to explain the empirical results that memorizing data-specific information can enhance generalization, particularly in the context of long-tailed data.
>
> In contrast, our work refines these models by incorporating the data-specific information with heterogeneous covariance. We theoretically prove that neural networks can leverage the implicit features learned from data-specific information to improve classification performance on long-tailed data. Our theory greatly extends the traditional understanding of overfitting.
>
>
> **Calculation of $||\mathbf{A}_i^\top\mathbf{A}_j||_F$**:
> To estimate the quantity $\mathbf{A}_i$ for real-world datasets, we decompose the intra-class data covariance matrix.
>
> Recall that in our data model, the covariance matrix of a data sample $(\mathbf{x},y)$ within class $i$ is $\boldsymbol{\Sigma}_i = \mathbf{A}_i\mathbf{A}_i^\top$.
> To estimate $\mathbf{A}_i$ for real-world datasets, we first compute the sample covariance matrix:
> \begin{align}
> \tilde{\boldsymbol{\Sigma}}_i = \frac{1}{|\mathcal{S}_i|-1}\sum\_{(\mathbf{x},y)\in\mathcal{S}_i} (\mathbf{x} -\bar{\mathbf{x}})(\mathbf{x} - \bar{\mathbf{x}})^\top,
> \end{align}
> where $|\mathcal{S}_i|$ is the size of class $i$ in the training dataset and $\bar{\mathbf{x}}$ is the sample mean.
>
> Next, we perform eigendecomposition of the sample covariance matrices $\tilde{\boldsymbol{\Sigma}}_i = \mathbf{Q}_i \boldsymbol{\Lambda}_i\mathbf{Q}_i^\top$,
> where $\mathbf{Q}_i$ contains the eigenvectors and $\boldsymbol{\Lambda}_i$ is the diagonal matrix of eigenvalues.
>
> We then estimate $\mathbf{A}_i$ as $\tilde{\mathbf{A}}_i = \mathbf{Q}_i\boldsymbol{\Lambda}_i^{1/2}$.
> Using the estimated $\tilde{\mathbf{A}}_i$ for each class, we can estimate $||\mathbf{A}_i^\top\mathbf{A}_j||_F$.
>
> We will add this detailed explanation in the appendix.
>
> [1] Allen-Zhu, Zeyuan, and Yuanzhi Li. "Feature purification: How adversarial training performs robust deep learning." IEEE Annual Symposium on Foundations of Computer Science, 2022.
>
> [2] Wu, Tong, et al. "Adversarial robustness under long-tailed distribution." Proceedings of the IEEE/CVF conference on computer vision and pattern recognition, 2021.
>
> [3] Cao, Yuan, et al. "Benign overfitting in two-layer convolutional neural networks." Advances in neural information processing systems, 2022: 25237-25250.
>
> [4] Kou, Yiwen, et al. "Benign overfitting in two-layer relu convolutional neural networks." International conference on machine learning, 2023.
>
> [5] Shang, Shuning, et al. "Initialization Matters: On the Benign Overfitting of Two-Layer ReLU CNN with Fully Trainable Layers." arXiv preprint arXiv:2410.19139 (2024).
>
> [6] Karhadkar, Kedar, et al. "Benign overfitting in leaky relu networks with moderate input dimension." Advances in Neural Information Processing Systems, 2024.
>
> [7] George, Erin, et al. "Training shallow ReLU networks on noisy data using hinge loss: when do we overfit and is it benign?." Advances in Neural Information Processing Systems, 2023.
>
> [8] Allen-Zhu, Zeyuan, and Yuanzhi Li. "Towards understanding ensemble, knowledge distillation and self-distillation in deep learning." arXiv preprint arXiv:2012.09816.

---

### Official Review · Reviewer_NBdR · 2025-03-10

**Overall Recommendation:** 3

**Summary:**

This paper establishes an enhanced feature-noise model by considering class-dependent heterogeneous noise across classes. Under this model, the paper demonstrates how memorization of long-tailed data boosts model performance. Additionally, it provides a phase transition of test error between benign and harmful overfitting.

**Claims And Evidence:**

Yes, the analysis is clear, rigorous, and thorough. Additionally, experiments are provided for further support.

**Essential References Not Discussed:**

No.

**Experimental Designs Or Analyses:**

Yes, I checked the experimental design section.

**Methods And Evaluation Criteria:**

Yes, the model is more reasonable and closer to real-world applications than previous work. The experiment covers both synthetic data generated from the model and real-world datasets.

**Other Comments Or Suggestions:**

No.

**Other Strengths And Weaknesses:**

Strength:
1. Novel Generalization: The paper extends the feature-noise model to class-dependent heterogeneous noise, making it more applicable to real-world scenarios.
2. Theoretical Insights: It provides a rigorous analysis demonstrating how memorization of long-tailed data boosts model performance.

Weaknesses:
1. Definition of Long-Tailed Data: The conditioning of the L-long-tailed data distribution on the neural network model at time step T may introduce a recursive or non-intuitive aspect. It is unclear why conditioning on W_T is justified when no new data is generated after optimization begins. Please refer to questions for detail.

**Questions For Authors:**

I have questions about the definition of long-tailed data. The L-long-tailed data distribution (Definition 1) is defined by conditioning on the neural network model at time step T. I feel there might be an intrinsic tricky part, like a recursive definition, because the training data is generated first, and the neural network parameters are then optimized using this data. It seems odd to condition on W_T because no new data is generated after the optimization process starts. This is different from using SGD, where new data is introduced in each iteration. It would make more sense to me if a long-tailed data subset of the entire training dataset were defined independently of the distribution, and then this subset’s specific role throughout the training dynamics were analyzed. Please correct me if I have any misunderstanding.

**Relation To Broader Scientific Literature:**

The generalization to heterogeneous data is good. For me, the interesting part is the results for long-tailed data, where the authors show that the classification accuracy of long-tailed data increases with the noise correlation ratio. This provides a theoretical explanation for the empirical observation of the necessity of long-tailed data in the training dataset.

**Theoretical Claims:**

I didn't check the correctness of the proofs, but I am familiar with the proof technique, and the results look reasonable to me. So, it is very likely to be correct.

---

> ### Author Rebuttal · Authors · 2025-04-01
>
> We sincerely appreciate the reviewer for recognizing our contributions and for the insightful comments.
> Below, we address your concerns in detail:
>
> **Clarification of Long-Tailed Data Definition**:
> Our definition is not recursive but instead relies on a fixed value of $T$, with no new data being introduced post-training, nor does it alter the training process.  Instead, we use the final model $\mathbf{W}^{(T)}$
> —obtained after a sufficient number of optimization steps $T$—to assess the long-tailedness of individual samples.
>
> Classical definitions of long-tailedness often hinge on simple one-dimensional statistics (e.g., sample counts or Zipf distribution). However, such metrics may fail to capture nuances of high-dimensional data (e.g., 3\*32\*32 images in CIFAR).
> For instance, for Sub-Gaussian data, norms alone do not always capture which samples are “rare” or “hard” to learn due to concentration effects.
>
> To address this, we draw inspiration from prior work that uses trained models to identify "long-tailed" samples.
> For example, Feldman and Zhang [1] evaluated long-tailedness by measuring the accuracy changes when training with or without a specific data sample.
> Similarly, Garg et al. [2] computed a curvature-based measure using trained models.
>
> In a similar manner, our definition leverages the trained model $\mathbf{W}^{(T)}$ to map high-dimensional samples to a one-dimensional metric, allowing us to identify long-tailed data points from the perspective of the network’s learned representation.
>
> We hope this explanation clarifies that conditioning on $\mathbf{W}^{(T)}$
> does not introduce a recursive definition and is instead a post-hoc method for characterizing long-tailed data distributions in high-dimensional settings.
>
> Thank you again for your feedback; please let us know if further elaboration is needed.
>
> [1] Feldman, Vitaly, and Chiyuan Zhang. "What neural networks memorize and why: Discovering the long tail via influence estimation." Advances in Neural Information Processing Systems, 2020.
>
> [2] Garg, Isha, Deepak Ravikumar, and Kaushik Roy. "Memorization Through the Lens of Curvature of Loss Function Around Samples." International Conference on Machine Learning, 2024.

---

### Official Review · Reviewer_pfPE · 2025-03-13

**Overall Recommendation:** 3

**Summary:**

This paper explores overfitting in neural networks, challenging the common belief that it harms generalization. Specifically, the authors argue that certain forms of overfitting can be benign and demonstrate—both mathematically and empirically—that benign overfitting can enhance performance on long-tailed data.

**Claims And Evidence:**

Yes

**Essential References Not Discussed:**

None that I know of

**Experimental Designs Or Analyses:**

Yes

**Methods And Evaluation Criteria:**

Yes. But analysis on more complex dataset such as ImageNet would further strengthen the claim made by the paper.

**Other Comments Or Suggestions:**

None found

**Other Strengths And Weaknesses:**

## Strengths
- The paper is well-written and easy to follow.
- It presents a novel perspective on overfitting, challenging conventional wisdom in the field.
- The claims are rigorously supported through both mathematical analysis and empirical validation, strengthening the arguments.

## Weaknesses
- The experimental section could be strengthened by evaluating more complex datasets, such as ImageNet, rather than relying solely on MNIST and CIFAR-10.

**Questions For Authors:**

One of the major concern is regarding the dataset used for evaluation. Is there any reason for not performing experiments on more complex dataset?

**Relation To Broader Scientific Literature:**

The paper is trying to explore the phenomenon of overfitting where it helps the model's performance. This is quite broad topic and can be applied to many research area.

**Theoretical Claims:**

No

---

> ### Author Rebuttal · Authors · 2025-04-01
>
> We sincerely appreciate the reviewer for recognizing our contributions and for the insightful comments. Below, we address your concerns in detail:
>
> **Clarification on the Choice of Datasets**: We would clarify the reason for experiments on MNIST.
>  - While MNIST may be considered simpler in comparison to other datasets, it remains a highly relevant choice for studying long-tailed data distributions. The long-tailed examples in MNIST—often corresponding to poorly written digits—offer clear physical interpretability [1].
>
> - Moreover, existing theoretical analyses in feature learning often rely on simplified synthetic data models (e.g., combining features with class-independent sub-Gaussian noise). Unlike those simpler models, our approach incorporates realistic intra-class covariance structures, similar to those in MNIST, providing new insights into the mechanism behind benign overfitting.
>
> **Applicability to More Complex Datasets**: We understand your concern about the applicability of our findings to larger and more complex datasets.
> To address this, we extended our experiments to CIFAR-100, a more challenging dataset, using a deeper network architecture, ResNet-101. We replicate the experimental settings from Figure 7 and present the updated results below:
>
> | Remove percentage | 0% | 20% | 40%| 60%| 80%|
> |----------|----------|----------|----------|----------|----------|
> |Remove images with high influence scores | 66.48% | 64.77% | 55.34% | 46.53% | 34.38%|
> |Remove images with low influence scores | 66.48% | 65.52% | 60.52% | 55.34% | 38.20% |
>
> These results further validate our theoretical predictions (particularly Statement 2(b) in Theorem 1). As shown, removing high-score samples results in a more significant accuracy drop compared to removing low-score samples. This trend aligns with our theoretical expectations, demonstrating that our framework applies to more challenging datasets like CIFAR-100, as well as deeper network architectures like ResNet-101.
>
> Due to time constraints, we plan to include additional experiments on Tiny ImageNet in an updated version of our work.
>
> We hope that these extended experiments clarify the broader applicability of our results to more complex datasets.
> Thank you again for your valuable comment.
>
> [1] Feldman, Vitaly, and Chiyuan Zhang. "What neural networks memorize and why: Discovering the long tail via influence estimation." Advances in Neural Information Processing Systems 33 (2020): 2881-2891.

---

### Official Review · Reviewer_SW6r · 2025-03-20

**Overall Recommendation:** 3

**Summary:**

This work investigates the phenomenon of benign overfitting in two-layer neural networks by rdiscussing a  feature-noise model. The authors introduce a class-dependent heterogeneous noise model to attempt to explain why neural nets can leverage long-tailed data distributions for generalization. The paper claim to provide theoretical results demonstrating a phase transition between benign and harmful overfitting and supports these findings with empirical evaluations on synthetic and real-world datasets. The authors argue that neural nets can learn implicit features from data-specific noise, which enhances classification accuracy, particularly for long-tailed distributions.

**Claims And Evidence:**

The central claim of the paper is that neural networks can use data-specific noise as an implicit feature to improve classification performance. While the theoretical results are rigorous, the paper’s framing of memorization is somewhat misleading. To me, the so-called "memorization of noise" is, in reality, learning from well-separated covariances rather than pure memorization of useless noise. The experimental results generally align with the theoretical findings, but the paper lacks a clear discussion on why its proposed model is the right one for studying benign overfitting in practical scenarios.

**Essential References Not Discussed:**

he paper should include more discussion of recent work on implicit regularization and the role of covariance structures in deep learning. Some relevant studies on feature learning dynamics in deep networks and their relation to benign overfitting are missing. Additionally, work on the role of batch normalization and other architectural choices in mitigating harmful overfitting should be considered.

On the top of my head, i can thnkg of work such as Srebro's

The Implicit Bias of Gradient Descent on Separable Data
Daniel Soudry, Elad Hoffer, Mor Shpigel Nacson, Suriya Gunasekar, Nathan Srebro; 19(70):1−57, 2018.

Or

The Double-Edged Sword of Implicit Bias: Generalization vs. Robustness in ReLU Networks

**Experimental Designs Or Analyses:**

This is a theory paper, so this kind of question seems ill-defined. The author do perform experiements, and the experimental setup is generally well-structured, with evaluations on both synthetic and real-world datasets (MNIST, CIFAR-10), even though i do not really count MNIST as meaningful. However, the key illustrative figures are unclear, particularly regarding the interpretation of "noise" and how it contributes to learning. It would be helpful to clarify how increasing the noise strength without a signal component affects performance and whether the conclusions hold for different model architectures.

**Methods And Evaluation Criteria:**

he authors use a combination of theoretical analysis and experimental validation. The theoretical methods appear sound and build upon prior work, extending it to the heterogeneous noise setting. However, the experimental evaluation could be improved by including comparisons with alternative models of benign overfitting, rather than focusing exclusively on their own framework. The evaluation criteria for success are reasonable but could be more explicitly tied to real-world applications (CIFAR is ok, but MNIST does not really count in myopinion).

**Other Comments Or Suggestions:**

Clarify what is meant by "memorizing noise"—if the noise contains label information, then it is not purely noise but an implicit feature.

Improve the illustrative figures to make it clearer how noise and signal interact in the proposed model.

Include a discussion on why this specific refinement of the feature-noise model is preferable to other potential explanations for benign overfitting.

Compare results against other recent theories of benign overfitting, such as those related to implicit regularization.

**Other Strengths And Weaknesses:**

Strengths: The paper presents an interesting refinement of existing models and contributes new theoretical insights into overfitting in long-tailed distributions. The analysis of singular values and covariance structures is a valuable addition.

Weaknesses: The framing of "memorization of noise" is somewhat misleading, as what is actually observed is learning from informative covariances. The motivation for the specific model chosen could be stronger, and the empirical results would benefit from comparisons to alternative benign overfitting models.

**Questions For Authors:**

Can you clarify why you refer to learning from structured noise as "memorization" when it appears to be leveraging class-dependent covariance information?

How would your results change if the noise distribution was not sub-Gaussian but had a different structure?

Would your findings hold in deeper networks beyond two-layer models, and how would they interact with batch normalization or dropout?

Could you include comparisons with other models of benign overfitting to contextualize your results?

**Relation To Broader Scientific Literature:**

The paper builds upon prior work on benign overfitting and feature-noise models, referencing key studies such as those by Kou et al. (2023) and Feldman & Zhang (2020). However, it does not sufficiently engage with alternative explanations for benign overfitting, such as implicit regularization effects in overparameterized models. These are classical and widely accepted. More discussion on the relationship between this work and theories of representation learning would strengthen its contribution.

**Theoretical Claims:**

The theoretical results appear correct and are well-supported by the provided proofs. The use of singular value distributions and covariance matrix properties seems to be an extension of previous work. However, the claim that memorizing noise is beneficial should be reconsidered; what is actually being demonstrated is that certain types of structured noise contain class-relevant information, making it part of the learnable signal rather than harmful noise.

---

> ### Author Rebuttal · Authors · 2025-04-01
>
> We sincerely appreciate the reviewer for recognizing our contributions and for the insightful comments. We will answer your questions in the following.
>
> **Discussion on Datasets**:
> Thanks for your comment. Please see the reply to Reviewer pfPE due to the space limit.
>
> **Clarification of Noise Memorization**:
>     Typically, memorization (overfitting) of a data sample is defined by achieving near-optimal training loss on that sample [1], a definition commonly adopted in the literature.
> Furthermore, the term 'memorization of noise' originates from feature learning studies [2,3], where it refers to situations in which classification relies heavily on data-specific information rather than the explicit feature.
> However, we agree that the term "noise" might be misleading, especially when it implicitly carries label-related signals.
> We will revise the terminology throughout the paper, replacing 'noise' with 'data-specific information' and explicitly clarifying its class-dependent nature.
>
> **Clarification of the Illustrative Figure**:
> In Figure 2, we clarify the feature strength as $||\mathbf{u}_i||_2$ and noise strength as $||\mathbf{A}_i^\top\mathbf{A}_i||_F$. Our paper includes settings and explanations for each figure, and we are happy to provide additional clarifications for any other figures that may require further elaboration.
>
> **Justification of the Proposed Model**:
> Our refinement is supported by two key observations.
> First, from the t-SNE visualization in Figure 1, we observe that different classes exhibit distinct intra-class distributions.
> Second, Figure 6 further verifies this by showing that real-world datasets, such as MNIST, have higher intra-class squared Frobenius norms compared to their inter-class counterparts.
>
> **Comparison with Other Benign Overfitting Models**:
> Due to the space limit, for a comparison with other models of benign overfitting, please see 'Comparison with Other Noise Models' for reviewer 5giJ.
>
> **Discussion on Implicit Bias/Regularization**:
> We will include an overview of studies and comparisons of implicit bias/regularization in the appendix.
> However, we would like to emphasize that our work is less relevant to implicit bias and regularization research.
> Implicit regularization is a broad research area focused on identifying high-level regularization effects of optimization algorithms, such as *max-margin, low-rank, and flat biases*, which display good generalization properties.
> In contrast, our paper is more aligned with research on feature learning, which detailedly characterizes how the *features are learned during training and how these learned features contribute to generalization*.
>
> **Discussion on Noise Structure**:
> We appreciate your comment regarding the applicability of sub-Gaussian distributions.
> Sub-Gaussian distributions are widely applicable in practice, as they encompass a broad range of real-world data distributions (e.g., ImageNet). Specifically, many natural and empirical data distributions are bounded, and such bounded distributions inherently exhibit sub-Gaussian characteristics. While other noise structures may exhibit different properties, sub-Gaussian distributions provide a strong and general framework that captures the behavior of most real-world data, making them a reasonable assumption in many contexts. Exploring the impact of alternative noise structures would undoubtedly be an interesting direction for future work.
>
> **Extension to Deeper Neural Networks and Modern Techniques**:
> From a theoretical perspective, analyzing neural networks with more than two layers presents significant challenges due to the non-convex and non-smooth nature of the loss function, as well as the complex interdependencies of parameters across layers.
> From an empirical perspective, we found that the main conclusions of our two-layer neural network analysis generally extend to deeper architectures.
> In our experiments with CIFAR-10 and CIFAR-100, we employ ResNet architectures with batch normalization, demonstrating that our theoretical predictions regarding covariance structure are applicable beyond simple two-layer models.
>
> [1] Cheng, Chen, John Duchi, and Rohith Kuditipudi. "Memorize to generalize: on the necessity of interpolation in high dimensional linear regression." Conference on Learning Theory, 2022.
>
> [2] Allen-Zhu, Zeyuan, and Yuanzhi Li. "Towards understanding ensemble, knowledge distillation and self-distillation in deep learning." arXiv preprint arXiv:2012.09816.
>
> [3] Cao, Yuan, et al. "Benign overfitting in two-layer convolutional neural networks." Advances in neural information processing systems, 2022.

---

### Decision · Program_Chairs · 2025-05-01

**Decision:**

Accept (poster)

**Comment:**

This paper investigates benign overfitting of two-layer neural networks by introducing a new concept of class-dependent heterogeneous noise and shows the signal-to-noise ratio threshold between benign-overfitting and harmful-overfitting. Through the analysis, they show that the class dependent noise enhances classification accuracy, particularly for long-tailed distributions.

The paper is well written and easy to follow. The idea of class-dependent heterogeneous noise is novel. This perspective mitigates the gap between theory and practice. The authors conducted solid theoretical analyses. Indeed, the reviewers did not raise a critical theoretical fault.

 As a summary, this paper can be accepted as a solid theoretical paper that contributes to the series of papers on benign overfitting.